# Associations between falls and other serious adverse events and antihypertensive medication in individuals with dementia: An observational cohort study

Takeshi Fujiwara[1,2], Constantinos Koshiaris[1,3], Ting Cai[1], Ariel Wang[1], Joseph Lee[1], Sarah Lay-Flurrie[1], Amitava Banerjee[4,5], Andrew Clegg[6], Rupert A. Payne[7,8], Subhashisa Swain[1], Margaret Ogden[9], Satoshi Hoshide[2], Kazuomi Kario[2], F. D. Richard Hobbs[1], Richard J. McManus[1,10], James P. Sheppard[1]*

1 Nuffield Department of Primary Care Health Sciences, University of Oxford, Oxford, United Kingdom, 2 Division of Cardiovascular Medicine, Department of Medicine, Jichi Medical University School of Medicine, Shimotsuke, Japan, 3 Department of Primary Care and Population Health, University of Nicosia Medical School, Nicosia, Cyprus, 4 Institute of Health Informatics, University College London, London, United Kingdom, 5 Department of Cardiology, Barts Health NHS Trust, London, United Kingdom, 6 Academic Unit for Ageing & Stroke Research, University of Leeds, Bradford Teaching Hospitals NHS Foundation Trust, Bradford, United Kingdom, 7 Exeter Collaboration for Academic Primary Care, University of Exeter, Exeter, United Kingdom, 8 Department of Health and Community Sciences, University of Exeter Medical School, Exeter, United Kingdom, 9 Patient and Public Involvement Contributor, 10 Brighton and Sussex Medical School, Brighton, East Sussex, United Kingdom

* james.sheppard@phc.ox.ac.uk

## Abstract

### Background

The balance of benefits and risks associated with lowering blood pressure levels in individuals with dementia remains controversial with a lack of evidence for possible harms associated with antihypertensive treatment. We examined the association between antihypertensive medication and serious adverse events in individuals with dementia compared to those without dementia.

### Methods and findings

This was a retrospective analysis using nationally representative UK general practice population between 1998 and 2018, from electronic health records (Clinical Practice Research Datalink, CPRD, GOLD). Eligible individuals were aged ≥40 years, with a systolic blood pressure 130–179 mmHg, and not previously prescribed antihypertensive treatment. The diagnosis of dementia was based on clinical codes in the electronic health record. Individuals were allocated to the exposure group if they were prescribed at least one antihypertensive medication during a 12-month exposure period. Those who were not prescribed any antihypertensive medication during the exposure period were allocated to the control group. The primary outcome was the first hospitalisation or death from a fall within 10 years of the follow-up period.

which permits unrestricted use, distribution, and reproduction in any medium, provided the original author and source are credited.

**Data availability statement:** Data were obtained via a CPRD institutional licence. Data cannot be shared publicly by the authors because of restrictions placed on its use by the data controller (the Clinical Practice Research Datalink [CPRD]). These require individuals needing access to the data to be specified in a protocol approved by the CPRD Independent Scientific Advisory Committee. The protocol must be included as part of a data sharing agreement between the data controller (CPRD) and the data processors (the authors). Data are available from the CPRD (contact via enquiries@cprd.com) for researchers who meet the criteria for access to these confidential data. The Hospital Episode Statistics data used in this analysis are re-used with permission from NHS Digital (contact via enquiries@nhsdigital.nhs.uk) who retain the copyright for that data. Complete code lists for variables used in this analysis can be found at https://github.com/jamessheppard48/STRATIFY-BP/tree/Causal-inference-project.

**Funding:** TF received funding from the SENSHIN Medical Research Foundation. JPS and CK receive funding from the Wellcome Trust/Royal Society via a Sir Henry Dale Fellowship (ref: 211182/Z/18/Z). This research was funded in part, by the Wellcome Trust (ref: 211182/Z/18/Z). FDRH is part supported by the NIHR Applied Research Collaboration (ARC) Oxford Thames Valley. RJM is supported by NIHR Oxford and Thames Valley Applied Research Consortium. The funders had no role in study design, data collection and analysis, decision to publish, or preparation of the manuscript.

**Competing interests:** The authors state that they have no potential conflicts of interest.

**Abbreviations:** aHR, adjusted hazard ratio; BMI, body mass index; BP, blood pressure; CI, confidence interval; CPRD, Clinical Practice Research Datalink; CVD, cardiovascular disease; HDL, high-density lipoprotein; RR, relative risks; SMD, standardised mean difference.

Secondary outcomes were first hospitalisation or death from hypotension, syncope, and fracture. In a population of 1,219,732 individuals, 23,510 had dementia. Antihypertensive medications were newly prescribed in 4,062/23,510 (17.3%) individuals with dementia and 142,385/1,196,222 (11.9%) individuals without dementia in the 12-month exposure period. In the primary analyses, which adjusted for the propensity score and a previous history of the outcome of interest, antihypertensive treatments were associated with a small increased risk of hospitalisation or death from falls (adjusted hazard ratio [aHR] 1.15, 95% confidence interval [CI] 1.08, 1.22), hypotension (aHR 1.51, 95%CI 1.29, 1.78), syncope (aHR 1.34, 95%CI 1.11, 1.61), but not fracture (aHR 1.05, 95%CI 0.96, 1.15), in individuals with dementia. These findings were consistent across different analytic approaches, including multivariable adjustment, propensity score matching, and inverse probability treatment weighting. In individuals without dementia, the association between antihypertensive treatment and serious adverse events was similar, with a small increased risk of hospitalisation or death from falls (aHR 1.07, 95%CI 1.05, 1.10). However, the absolute fall risk associated with antihypertensive treatment was significantly higher in individuals with dementia (47 per 10,000 individuals per year, 95%CI 26, 70) compared to those without (14 per 10,000 individuals per year, 95%CI 10, 18). The absolute risks of hypotension and syncope with antihypertensive treatment were also higher in the individuals with dementia compared to those without. The main limitation is the possibility of unmeasured confounding, and heterogeneity in dementia diagnoses based on coded entries in the electronic health record.

## Conclusions

Antihypertensive treatment was associated with increased risk of serious adverse events in individuals with and without dementia, however, the absolute risk of harm was more than double in individuals with dementia. These data suggest that clinicians, patients, and their carers should consider these risks before starting new antihypertensive medications, particularly in the context of dementia.

### Author summary
#### Why was this study done?

- Antihypertensive drugs have been shown to reduce cardiovascular disease morbidity and mortality among all age groups.

- Previous studies have demonstrated that antihypertensive treatment is associated with increased risk of serious adverse events, especially in older people and those with severe frailty.

- However, there is a lack of firm evidence regarding the harm associated with antihypertensive treatment in individuals with dementia.

PLOS Medicine

**What did the researchers do and find?**

- This was a retrospective analysis using nationally representative UK general practice population between 1998 and 2018, from electronic health records.

- In a total of 1,219,732 individuals (23,510 with dementia), antihypertensive treatment was associated with an increased risk of hospitalisation or death from falls, hypotension, and syncope in both individuals with and without dementia.

- The absolute risk of harm with antihypertensive treatment was significantly higher in individuals with dementia.

**What do these findings mean?**

- These findings suggest that clinicians, patients, and their carers should consider the risk of serious adverse events when starting new antihypertensive medications for individuals with dementia.

- The limitation is the potential for unmeasured confounding, and heterogeneity in dementia diagnoses based on coded diagnoses in the electronic health record.

## Introduction

Pharmacological treatments to lower blood pressure (BP) have been shown to reduce morbidity and mortality from cardiovascular disease (CVD) [1,2]. Treatment benefits are broadly consistent among all age groups [3]. Indeed, in the last decade, the beneficial effects of strict BP control for reducing CVD event risk in active older people have been demonstrated [4,5]. Recent evidence suggests that individuals with frailty may still benefit from antihypertensive therapy [6,7], and a systematic review has shown that even frail older adults may derive cardiovascular benefit from BP lowering [8]. Treating hypertension in older people is an important task for primary care physicians. However, caution should be exercised, as older people are at increased risk of serious adverse events, especially in those with progression of frailty [9]. Recent hypertension guidelines recommend that careful assessment of the balance of benefits and harms from antihypertensive treatment is essential in older people [10–12], although these recommendations are primarily based on expert opinion and low-grade evidence, because there are few trials of antihypertensive treatment in older people, particularly those with frailty and dementia.

Dementia has a major and increasing global health burden with numbers affected expected to nearly triple from 57 million in 2019 to more than 153 million worldwide by 2050 [13]. Hypertension, one of the modifiable risk factors for dementia [14], is common in individuals with dementia [15], but there has been controversy about the balance of benefits and risks associated with lowering BP levels in individuals with dementia. Recent evidence from a randomised controlled trial demonstrated that intensive BP reduction was effective in lowering the risk of all-cause dementia among patients with hypertension, suggesting a potential preventive role for antihypertensive treatment [16]. Some studies showed that antihypertensive medications increased cerebral blood flow in individuals with dementia [17,18], while another observational study demonstrated that low daytime systolic BP was associated with greater progression of cognitive decline in individuals with mild cognitive impairment or dementia among those treated with antihypertensive medications [19]. Dementia has been shown to be associated with increased risk of CVD events [20,21], but there is no direct evidence to date that antihypertensive medications could reduce CVD risks in individuals with dementia. Antihypertensive drug use in individuals with dementia may increase the risk of serious adverse events due to the presence of multimorbidity and polypharmacy [22,23]. In addition, individuals with dementia are more likely to be physically frail [24]. A discrete choice experiment suggested that clinicians favour altering or withdrawing unnecessary antihypertensive medications to decrease the risk of falls in individuals with dementia [25]. However, to date, there has been a lack of empirical evidence over the possible harms associated with antihypertensive treatment in individuals with dementia.

We therefore set out to determine whether antihypertensive treatment is associated with an increased risk of serious adverse events in individuals with dementia compared to those without.

## Methods

This study is reported using the REporting of studies Conducted using Observational Routinely-collected Data (RECORD) guideline (S1 Table). The Clinical Practice Research Datalink (CPRD) has global ethical approval for the use of anonymised electronic health records for research purposes, subject to approval of a study protocol by their Independent Scientific Advisory Committee. The protocol for this study was given prospective approval in February 2019 (ISAC protocol number 19_042) and is provided in S1 Protocol.

### Study setting

This was a retrospective observational cohort study. We used UK primary care electronic health record data held within CPRD GOLD (based on data from practices using Vision Care Electronic Health Record Software; Meddbase Software—Medical Management Systems, London, England). CPRD GOLD covers over 20 million individuals from 968 practices in the UK and included people are broadly representative of the UK population in the terms of age, sex, and ethnicity [26].

### Study population

Eligible individuals were those: (1) aged ≥40 years old; (2) with qualifying first systolic BP levels of between 130 and 179 mmHg [12] prior to the exposure period; (3) not having received any antihypertensives prior to the study start date; and (4) who were registered between 1st January 1998 and 31st December 2018 in CPRD GOLD. The exposure and follow-up periods were defined relative to each individual's cohort entry date, with a 1-year exposure assessment period followed by up to 10 years of follow-up. This design reflects a representative sample of the adult population across age groups and focuses on new users of antihypertensive treatment, allowing for a clearer assessment of treatment-related risks. A fixed follow-up duration was applied uniformly across individuals to ensure comparability of absolute risk estimates between exposure groups. The exclusion criteria were: (1) no record of BP measurement; and (2) qualifying systolic BP ≥ 180 mmHg, since those with a BP above this level were considered to require antihypertensive treatment prescription, regardless of serious adverse event risk [12]. Individual's characteristics were determined from information recorded at any point prior to the start of the follow-up period. Individuals exited the study on the study end date, when they transferred out of a registered CPRD practice, died, or experienced the specific outcome of interest (S1 Fig).

### Definitions of dementia

The diagnosis of dementia was based on clinical codes for dementia. We defined all-cause dementia using all pathological classifications, including Alzheimer's disease, vascular dementia, and other types of dementia (defined according to ICD-9 and ICD-10 codes listed S2 Table). Individuals with mild cognitive impairment were not included, because we could not consistently identify individuals with mild cognitive impairment within the dataset.

### Exposure

The exposure was prescription of any antihypertensive medication as defined in the British National Formulary (S3 Table). Individuals were allocated to the exposure group if they had been prescribed at least one antihypertensive medication during a 12-month period prior to the start date of the follow-up. Medications at baseline were defined by the most recent prescriptions prior to this start date. Those people did not have any antihypertensive medication prescribed during this period were allocated to the control group.

## Outcomes

The primary outcome was the first hospitalisation or death from a fall within 10 years of follow-up. Secondary outcomes were first hospitalisation or death from hypotension, syncope, or fracture. Outcomes were captured from ICD-9 and ICD-10 codes as the primary cause of admission in the Basic Inpatient Hospital Episode Statistics and primary cause of death on the death certificates from the Office for National Statistics.

## Multiple imputation for missing variables

The percentages of missing variable in the present study were as follow: body mass index (BMI) 15.2%, ethnicity 62.7%, smoking status 4.8%, alcohol consumption 14.8%, deprivation 0.08%, total cholesterol 53.9%, and high-density lipoprotein (HDL) cholesterol 65.5%. In total 110,666 out of 1,219,732 (9.1%) were complete cases. Missing data were imputed using multiple imputations with chained equations, using the "mice" package in R. We created 10 imputed datasets for both individuals with and without dementia groups each and analysed treatment effects in each imputation dataset separately [27]. These estimates and their standard errors were combined using Rubin's rules [28]. The outcome of interest from each analysis was included in separate imputation models. We also performed a sensitivity analysis on the subset of complete cases.

## Covariates

Covariates included demographics (age, gender, ethnicity and index of multiple deprivation), clinical characteristics (systolic and diastolic BP, BMI, total cholesterol, HDL cholesterol, smoking status and alcohol consumption), past medical history (stroke, transient ischaemic attack, myocardial infarction, heart failure, peripheral vascular disease, coronary artery bypass graft, angina, chronic kidney disease, diabetes, atrial fibrillation and cancer), other prescribed medications (statins, anti-thrombotics [including both antiplatelets and anticoagulants], opioids, hypnotics/anxiolytics, antidepressants and anti-cholinergic medications), frailty (using the validated electronic frailty index, which combines 36 deficits) [29], primary CVD risk (based on the QRisk2 score, where the QRisk2 score was unavailable, it was replaced with the population mean) [30], and previous history of the outcome of interest. These covariates were selected *a priori* based on clinical treatment guidelines and expert opinion [12].

Deprivation was assessed using the English Index of Multiple Deprivation 2015 (https://www.gov.uk/government/statistics/english-indices-of-deprivation-2015). The Index of Multiple Deprivation 2015 is the official measure of relative deprivation and provides a comprehensive overview of socioeconomic deprivation across various regions in England [31]. In the present study, the Index of Multiple Deprivation scores were divided into quintiles, with the Index of Multiple Deprivation score of five indicating individuals in the highest quintile of deprivation (most deprived).

## Propensity score generation

To account for differences in baseline characteristics between the two groups, we conducted propensity score analyses separately within individuals with and without dementia. In each of these two groups, we generated propensity scores in each imputation dataset using multivariable logistic regression with prescription of antihypertensive medication as the outcome, and all baseline covariates included (listed in the previous section and in S4 and S5 Tables). Continuous variables were categorised to account for nonlinear associations with the outcome (the use of splines and fractional polynomials was explored but led to model convergence issues). For the matched analysis, a 1:1 nearest neighbour matching approach was used with a calliper restriction of 0.2. Standardised mean differences (SMDs) were estimated for all baseline covariates before and after matching to assess pre-matching and post-matching covariate balance. We considered an SMD of <0.1 for a covariate as indication of acceptable balance [32].

## Main analysis

The cumulative incidence of serious adverse events was estimated by the Kaplan–Meier method. For the primary analysis, propensity scores and a previous history of the outcome of interest were adjusted in Cox regression models to examine the association between antihypertensive treatment and serious adverse events. For secondary analyses, we used: (1) Cox regression models which were adjusted for the same variables included in the propensity score models; (2) propensity score matched multivariate adjusted Cox regression models; and (3) inverse probability treatment weighting model. Those three models also included a previous history of the outcome of interest as an adjustment variable. Inverse probability treatment weighting was calculated as the inverse of the propensity score of individuals in the exposure group and the inverse (1—propensity score) for those in the control group [33,34]. The inverse probability treatment weighting model was applied to generate a weighted cohort [35]. These four methods were expected to provide similar results and were used to demonstrate the robustness of the results. All four models were fitted separately for individuals with and without dementia to estimate dementia-specific associations between antihypertensive treatment and each outcome. Schoenfeld residuals were examined to check the proportional-hazards assumption. Absolute risk differences were defined as the risk of each outcome in the population assuming exposure to antihypertensive treatment, minus the risk of each outcome assuming no exposure to antihypertensive treatment, using treatment effect estimates derived from the Cox regression models. This risk difference was estimated and reported as the number of events per 10,000 individuals treated per year [36], with 95% confidence intervals (CIs) obtained using 200 bootstrap iterations per imputation data set. Numbers needed to harm were calculated from the absolute risk difference.

## Subgroup and sensitivity analyses

Serious adverse event risks by number of antihypertensive drugs used were calculated for each outcome, by dementia status. The number of antihypertensive drugs were categorised into 0, 1, 2, or ≥3, and serious adverse event risks were calculated for each outcome in both groups using the number of antihypertensive drugs used zero as the reference category, using propensity score adjustment to control for confounding.

To assess whether there was a linear trend in the risk of adverse events across increasing numbers of antihypertensive medications, we additionally conducted a trend analysis by modelling the number of antihypertensive drugs as an ordered categorical variable in the Cox models. This approach allowed us to formally test for a linear increase in hazard with increasing medication count, after adjusting for imputed propensity scores and prior history of the outcome. Trend tests were conducted separately in individuals with and without dementia using pooled estimates from multiply imputed datasets.

During the study period, the use of diagnostic codes for dementia increased due to two factors: (1) the introduction of the NHS England national enhanced service for dementia diagnosis in 2013/14, and (2) the introduction of a dementia register in the Quality and Outcomes Framework incentivization scheme introduced in 2014/15. These factors may have affected the specificity of clinical codes for dementia, and the association between the risk of serious adverse events with antihypertensive medication and dementia status in this study. Therefore, sensitivity analyses were performed to assess whether the association between the risk of serious adverse events with antihypertensive medication and dementia status differed depending on whether the follow-up period started before or after April 1, 2014.

## Results

### Population characteristics

From a total of 16,071,111 registered individuals, 1,219,732 fulfilled the eligibility criteria (S2 Fig). Among them, 23,510 (1.9%) had dementia. In the 12-month exposure period, 4,062 (17.3%) of the 23,510 individuals with dementia and 142,385 (11.9%) of the 1,196,222 individuals without dementia started at least one antihypertensive medication,

respectively, and were included in the exposure groups. Before propensity score matching, there were differences in almost all baseline characteristics between individuals in the exposure group and those in the control group in both individuals with dementia and those without (Table 1). One-to-one matching by propensity score analysis resulted in 3,795 matched individuals with dementia and 131,199 matched those without between the exposure and control groups, respectively. After matching, the SMD for all variables included in propensity score was reduced to below 0.1, indicating effective matching (Tables 1 and S6).

To further assess the appropriateness of propensity score methods, we examined the distribution of propensity scores in individuals with and without dementia (S3 Fig).

### Primary outcome

During a median follow-up of 6.3 years (interquartile range 2.7, 10.0) years, a total of 8,314 individuals with dementia (35.4%) and 87,785 those without (7.3%) were hospitalised or died following a fall. The cumulative incidence of falls was higher in individuals with dementia compared to those without, and in the exposure group compared to the control group (Fig 1A). In the primary analyses using propensity score adjustment, antihypertensive treatment was associated with an increased risk of hospitalisation or death from falls in both individuals with dementia (adjusted hazard ratio [aHR] 1.15, 95% CI 1.08, 1.22) and in those without (aHR 1.07, 95%CI 1.05, 1.10) (Table 2). Analyses using multivariable adjustment, propensity score matching, and inverse probability treatment weighting showed similar results. While the interaction was statistically significant in three of the four analytical approaches, it was not significant in the inverse probability treatment weighting model. The absolute risk difference of falls with antihypertensive treatment was significantly higher in individuals with dementia (absolute risk difference 47 per 10,000 individuals per year, 95%CI 26, 70, equivalent to a number of needed to harm of 213 per year) compared to those without (absolute risk difference 14 per 10,000 individuals per year, 95%CI 10, 18, equivalent to a number of needed to harm of 714 per year) (Fig 2). The results were similar when analyses were undertaken using complete case only (S7 Table; S4 and S5 Figs).

### Secondary outcomes

In individuals with dementia, a total of 1,005 (4.3%) individuals experienced serious hypotension, 860 (3.7%) experienced syncope, and 4,125 (17.5%) individuals experienced a fracture, in each case warranting hospital admission, or associated with death. In those without dementia, 9,634 (0.8%) experienced serious hypotension, 13,766 (1.2%) experienced syncope, and 73,155 (6.1%) experienced a fracture. The cumulative incidence rates of these serious adverse events were higher in individuals with dementia and in the exposure group (Fig 1B–1D). Antihypertensive treatment was associated with an increased risk of hospitalisation or death from hypotension and syncope in both individuals with dementia and those without, but not from fracture: in individuals with dementia, hypotension (aHR 1.51, 95%CI 1.29, 1.78), syncope (aHR 1.34, 95%CI 1.11, 1.61), and fracture (aHR 1.05, 95%CI 0.96, 1.15); in individuals without dementia, hypotension (aHR 1.70, 95%CI 1.60, 1.79), syncope (aHR 1.25, 95%CI 1.18, 1.31), and fracture (aHR 1.00, 95%CI 0.98, 1.02) (Table 3). The absolute risk differences of hypotension and syncope were higher in individuals with dementia compared to those without (Fig 2). Similar results were observed when analyses were undertaken using complete cases only (S7 Table; S4 and S5 Figs).

### Subgroup analyses

In individuals both with and without dementia, the risks of falls and hypotension increased with increasing number of antihypertensive drugs (Table 4). There were no significant association between the risk of fracture and the number of antihypertensive drugs, irrespective of dementia diagnoses. The results were similar when analyses were undertaken using complete cases only (S8 Table).

**Table 1. Baseline characteristics of the study population before and after propensity score matching, with standardised mean differences between exposure and control groups.**

| | With dementia | | | | | | Without dementia | | | | | |
|---|---|---|---|---|---|---|---|---|---|---|---|---|
| | Before matching | | | After matching | | | Before matching | | | After matching | | |
| Characteristics | Antihypertensive prescription (−) | Antihypertensive prescription (+) | SMD | Antihypertensive prescription (−) | Antihypertensive prescription (+) | SMD | Antihypertensive prescription (−) | Antihypertensive prescription (+) | SMD | Antihypertensive prescription (−) | Antihypertensive prescription (+) | SMD |
| Number | 19,448 | 4,062 | | 3,795 | 3,795 | | 1,053,837 | 142,385 | | 131,199 | 131,199 | |
| Age, yrs | 75 [69, 81] | 76 [70,81] | 0.124 | 76 [70,82] | 76 [70,81] | 0.025 | 54 [46,63] | 61 [52,71] | 0.466 | 62 [53,71] | 61 [51,71] | 0.087 |
| Female, n (%) | 11,870 (61.0) | 2,503 (61.6) | 0.012 | 2,372 (62.5) | 2,358 (62.1) | 0.008 | 529,811 (50.3) | 69,547 (49.2) | 0.029 | 67,211 (51.2) | 64,563 (49.2) | 0.040 |
| Body mass index, kg/m² | 24.9 [22.4, 27.6] | 25.7 [23.0, 28.8] | 0.188 | 25.5 [22.8, 28.4] | 25.6 [23.0, 28.6] | 0.018 | 26.1 [23.4, 29.3] | 27.3 [24.3, 30.9] | 0.218 | 27.2 [24.3, 30.9] | 27.4 [24.4, 31.0] | 0.028 |
| Ethnicity | | | 0.098 | | | 0.036 | | | 0.107 | | | 0.014 |
| White ethnicity, n (%) | 19,037 (97.9) | 3,914 (96.4) | | 3,677 (96.9) | 3,667 (96.9) | | 1,003,556 (95.2) | 132,979 (93.4) | | 123,387 (94.0) | 123,008 (93.8) | |
| Black ethnicity, n (%) | 108 (0.6) | 54 (1.3) | | 34 (0.9) | 48 (1.3) | | 11,600 (1.1) | 3,434 (2.4) | | 2,818 (2.1) | 2,904 (2.2) | |
| South Asian ethnicity, n (%) | 123 (0.6) | 40 (1.0) | | 36 (0.9) | 35 (0.9) | | 15,749 (1.5) | 2,786 (2.0) | | 2,204 (1.7) | 2,427 (1.9) | |
| Other ethnicity, n (%) | 180 (0.9) | 54 (1.3) | | 48 (1.3) | 45 (1.2) | | 22,932 (2.2) | 3,186 (2.2) | | 2,790 (2.1) | 2,860 (2.2) | |
| Indices of multiple deprivation* | | | 0.024 | | | | | | 0.096 | | | 0.002 |
| Quintile 1, n (%) | 4,530 (23.3) | 957 (23.6) | | 878 (23.1) | 878 (23.1) | | 263,352 (25.0) | 30,936 (21.7) | | 28,797 (21.9) | 28,873 (22.0) | |
| Quintile 2, n (%) | 4,378 (22.5) | 911 (22.4) | | 852 (22.5) | 856 (22.6) | | 245,914 (23.3) | 32,388 (22.7) | | 29,985 (22.9) | 29,915 (22.8) | |
| Quintile 3, n (%) | 4,220 (21.7) | 874 (21.5) | | 838 (22.1) | 822 (21.7) | | 223,281 (21.2) | 30,490 (21.4) | | 27,953 (21.3) | 28,025 (21.4) | |
| Quintile 4, n (%) | 3,562 (18.3) | 718 (17.7) | | 668 (17.6) | 683 (18.0) | | 179,852 (17.1) | 26,427 (18.6) | | 24,324 (18.5) | 24,245 (18.5) | |
| Quintile 5, n (%) | 2,758 (14.2) | 602 (14.8) | | 559 (14.7) | 556 (14.7) | | 141,438 (13.4) | 22,144 (15.6) | | 20,140 (15.4) | 20,141 (15.4) | |
| Smoking status | | | 0.156 | | | 0.099 | | | 0.023 | | | 0.017 |
| Non-smoker, n (%) | 11,249 (57.8) | 2,189 (53.9) | | 2,113 (55.7) | 2,071 (54.6) | | 541,351 (51.4) | 68,212 (47.9) | | 64,372 (49.1) | 63,348 (48.3) | |
| Ex-smoker, n (%) | 5,386 (27.7) | 1,402 (34.5) | | 1,238 (32.6) | 1,275 (33.6) | | 271,859 (25.8) | 46,440 (32.6) | | 41,331 (31.5) | 41,716 (31.8) | |
| Current smoking status, n (%) | 2,813 (14.5) | 471 (11.6) | | 444 (11.7) | 449 (11.8) | | 240,627 (22.8) | 27,733 (19.5) | | 25,496 (19.4) | 26,135 (19.9) | |
| Alcohol consumption | | | 0.099 | | | | | | 0.158 | | | 0.018 |
| Non-drinker, n (%) | 3,861 (19.9) | 899 (22.1) | | 843 (22.2) | 845 (22.3) | | 176,531 (16.8) | 31,375 (22.0) | | 28,878 (22.0) | 28,307 (21.6) | |
| Trivial, n (%) | 4,608 (23.7) | 1,026 (25.3) | | 944 (24.9) | 942 (24.8) | | 341,500 (32.4) | 40,707 (28.6) | | 38,425 (29.3) | 37,997 (29.0) | |
| Light, n (%) | 6,660 (34.2) | 1,245 (30.6) | | 1,211 (31.9) | 1,177 (31.0) | | 178,941 (17.0) | 20,861 (14.7) | | 19,171 (14.6) | 19,483 (14.9) | |
| Moderate, n (%) | 2,818 (14.5) | 571 (14.1) | | 510 (13.4) | 527 (13.9) | | 140,237 (13.3) | 17,597 (12.4) | | 15,833 (12.1) | 16,342 (12.5) | |
| Heavy drinker, n (%) | 1,321 (6.8) | 298 (7.3) | | 268 (7.1) | 281 (7.4) | | 21,246 (2.0) | 3,239 (2.3) | | 2,901 (2.2) | 3,014 (2.3) | |
| Not reported, n (%) | 180 (0.9) | 23 (0.6) | | 19 (0.5) | 23 (0.6) | | 195,382 (18.5) | 28,606 (20.1) | | 25,991 (19.8) | 26,056 (19.9) | |
| QRisk2 score ≥10%, n (%) | 18,102 (93.1) | 3,989 (98.2) | 0.253 | 3,716 (97.9) | 3,722 (98.1) | 0.011 | 418,773 (39.7) | 110,353 (77.5) | 0.830 | 98,915 (75.4) | 99,336 (75.7) | 0.007 |
| Frailty status | | | 0.156 | | | 0.009 | | | 0.149 | | | 0.009 |
| Fit (eFI < 0.12) | 18,920 (97.3) | 3,826 (94.2) | | 3,603 (94.9) | 3,596 (94.8) | | 1,045,618 (99.2) | 138,492 (97.3) | | 128,243 (97.7) | 128,084 (97.6) | |
| Mildly frail (0.12 ≤ eFI < 0.24) | 227 (1.2) | 119 (2.9) | | 92 (2.4) | 95 (2.5) | | 3,737 (0.4) | 1,806 (1.3) | | 1,317 (1.0) | 1,428 (1.1) | |

*(Continued)*

| | With dementia | | | | | | Without dementia | | | | | |
| | Before matching | | | After matching | | | Before matching | | | After matching | | |
| | Antihypertensive prescription (−) | Antihypertensive prescription (+) | SMD | Antihypertensive prescription (−) | Antihypertensive prescription (+) | SMD | Antihypertensive prescription (−) | Antihypertensive prescription (+) | SMD | Antihypertensive prescription (−) | Antihypertensive prescription (+) | SMD |
|---|---|---|---|---|---|---|---|---|---|---|---|---|
| Moderately frail (0.24 ≤ eFI < 0.36) | 281 (1.4) | 108 (2.7) | | 91 (2.4) | 95 (2.5) | | 4,199 (0.4) | 1,943 (1.4) | | 1,522 (1.2) | 1,574 (1.2) | |
| Severely frail (0.36 ≤ eFI) | 20 (0.1) | 9 (0.2) | | 9 (0.2) | 9 (0.2) | | 283 (<0.1) | 144 (0.1) | | 117 (0.1) | 113 (0.1) | |
| Systolic blood pressure, mmHg | 142 [138, 154] | 150 [140, 160] | 0.425 | 150 [140, 160] | 150 [140, 160] | 0.017 | 140 [132, 148] | 150 [140, 160] | 0.741 | 150 [140, 160] | 150 [140, 160] | 0.023 |
| Diastolic blood pressure, mmHg | 80 [76, 88] | 82 [78, 90] | 0.257 | 82 [78, 90] | 82 [78, 90] | 0.042 | 82 [80, 90] | 89.0 [80, 97] | 0.477 | 90 [80, 94] | 88 [80, 97] | 0.096 |
| **Co-morbidities** | | | | | | | | | | | | |
| Stroke, n (%) | 604 (3.1) | 349 (8.6) | 0.235 | 305 (8.0) | 307 (8.1) | 0.002 | 11,403 (1.1) | 5,601 (3.9) | 0.183 | 4,457 (3.4) | 4,623 (3.5) | 0.007 |
| Transient ischaemic attack, n (%) | 310 (1.6) | 171 (4.2) | 0.156 | 145 (3.8) | 149 (3.9) | 0.005 | 5,173 (0.5) | 2,701 (1.9) | 0.130 | 2,151 (1.6) | 2,234 (1.7) | 0.005 |
| Myocardial infarction, n (%) | 299 (1.5) | 300 (7.4) | 0.286 | 198 (5.2) | 222 (5.9) | 0.028 | 5,318 (0.5) | 7,679 (5.4) | 0.292 | 3,694 (2.8) | 4,854 (3.7) | 0.050 |
| Heart failure, n (%) | 199 (1.0) | 138 (3.4) | 0.162 | 111 (2.9) | 114 (3.0) | 0.005 | 3,392 (0.3) | 3,581 (2.5) | 0.186 | 2,138 (1.6) | 2,478 (1.9) | 0.020 |
| Peripheral vascular disease, n (%) | 171 (0.9) | 101 (2.5) | 0.125 | 72 (1.9) | 81 (2.1) | 0.017 | 3,579 (0.3) | 1,766 (1.2) | 0.102 | 1,385 (1.1) | 1,456 (1.1) | 0.005 |
| Coronary artery bypass graft, n (%) | 55 (0.3) | 84 (2.1) | 0.166 | 45 (1.2) | 53 (1.4) | 0.019 | 993 (0.1) | 1,498 (1.1) | 0.127 | 753 (0.6) | 964 (0.7) | 0.020 |
| Angina, n (%) | 571 (2.9) | 489 (12.0) | 0.351 | 346 (9.1) | 394 (10.4) | 0.043 | 9,296 (0.9) | 9,716 (6.8) | 0.312 | 5,527 (4.2) | 6,758 (5.2) | 0.044 |
| Atrial fibrillation, n (%) | 652 (3.4) | 359 (8.8) | 0.231 | 302 (8.0) | 312 (8.2) | 0.010 | 10,614 (1.0) | 6,982 (4.9) | 0.232 | 5,407 (4.1) | 5,625 (4.3) | 0.008 |
| Diabetes mellitus, n (%) | 1,270 (6.5) | 601 (14.8) | 0.270 | 484 (12.8) | 516 (13.6) | 0.025 | 43,506 (4.1) | 20,070 (14.1) | 0.352 | 16,351 (12.5) | 16,693 (12.7) | 0.008 |
| Chronic kidney disease, n (%) | 241 (1.2) | 222 (5.5) | 0.236 | 140 (3.7) | 167 (4.4) | 0.036 | 7,324 (0.7) | 5,747 (4.0) | 0.221 | 3,653 (2.8) | 4,196 (3.2) | 0.024 |
| Cancer, n (%) | 1,173 (6.0) | 305 (7.5) | 0.059 | 259 (6.8) | 280 (7.4) | 0.022 | 36,095 (3.4) | 7,058 (5.0) | 0.077 | 6,273 (4.8) | 6,458 (4.9) | 0.007 |
| **Treatment prescription** | | | | | | | | | | | | |
| ACE inhibitors, n (%) | – | 1,385 (34.1) | – | – | 1,241 (32.7) | – | – | 58,068 (40.8) | – | – | 51,482 (39.2) | – |
| ARBs, n (%) | – | 365 (9.0) | – | – | 325 (8.6) | – | – | 14,359 (10.1) | – | – | 12,793 (9.8) | – |
| Calcium channel blockers, n (%) | – | 1,273 (31.3) | – | – | 1,179 (31.1) | – | – | 42,462 (29.8) | – | – | 38,262 (29.2) | – |
| Diuretics, n (%)† | – | 1,661 (40.9) | – | – | 1,599 (42.1) | – | – | 45,405 (31.9) | – | – | 42,674 (32.5) | – |
| Beta-blockers, n (%) | – | 1,434 (35.3) | – | – | 1,301 (34.3) | – | – | 50,866 (35.7) | – | – | 45,777 (34.9) | – |
| Alpha-blocker, n (%) | – | 232 (5.7) | – | – | 216 (5.7) | – | – | 6,512 (4.6) | – | – | 5,793 (4.4) | – |
| Other antihypertensives, n (%)‡ | – | 38 (0.9) | – | – | 35 (0.9) | – | – | 2,846 (2.0) | – | – | 2,715 (2.1) | – |
| Statins, n (%) | 1,570 (8.1) | 1,371 (33.8) | 0.665 | 1,058 (27.9) | 1,116 (29.4) | 0.034 | 66,052 (6.3) | 45,169 (31.7) | 0.686 | 32,291 (24.6) | 35,304 (26.9) | 0.053 |

*(Continued)*

**Table 1.** (Continued)

| | With dementia | | | | | | Without dementia | | | | | |
| | Before matching | | | After matching | | | Before matching | | | After matching | | |
| | Antihypertensive prescription (−) | Antihypertensive prescription (+) | SMD | Antihypertensive prescription (−) | Antihypertensive prescription (+) | SMD | Antihypertensive prescription (−) | Antihypertensive prescription (+) | SMD | Antihypertensive prescription (−) | Antihypertensive prescription (+) | SMD |
|---|---|---|---|---|---|---|---|---|---|---|---|---|
| Antiplatelets/anticoagulants, n (%) | 3,781 (19.4) | 1,962 (48.3) | 0.640 | 1,675 (44.1) | 1,711 (45.1) | 0.019 | 70,646 (6.7) | 44,019 (30.9) | 0.652 | 32,656 (24.9) | 34,967 (26.7) | 0.040 |
| Anticholinergics, n (%) | 2,699 (13.9) | 491 (12.1) | 0.053 | 440 (11.6) | 465 (12.3) | 0.020 | 99,544 (9.4) | 12,556 (8.8) | 0.022 | 12,060 (9.2) | 11,819 (9.0) | 0.006 |
| Antidepressants, n (%) | 3,925 (20.2) | 775 (19.1) | 0.028 | 720 (19.0) | 725 (19.1) | 0.003 | 191,826 (18.2) | 26,651 (18.7) | 0.013 | 26,125 (19.9) | 24,701 (18.8) | 0.027 |
| Hypotonic/anxiolytics, n (%) | 3,926 (20.2) | 740 (18.2) | 0.050 | 688 (18.1) | 697 (18.4) | 0.006 | 189,450 (18.0) | 24,705 (17.4) | 0.016 | 24,126 (18.4) | 23,030 (17.6) | 0.002 |
| Opioids, n (%) | 5,948 (30.6) | 1,174 (28.9) | 0.037 | 1,053 (27.7) | 1,099 (29.0) | 0.027 | 282,934 (26.8) | 40,448 (28.4) | 0.035 | 38,120 (29.1) | 37,205 (28.4) | 0.015 |

Data are median [interquartile range] or number (percentage). A SMD of <0.1 suggests adequate variable balance after propensity score matching.

*High deprivation indicates indices of multiple deprivation score of 5 (most deprived).

†Diuretics included thiazides and thiazide-like diuretics.

‡Other antihypertensives includes centrally acting antihypertensives, direct renin inhibitors, vasodilators, anti-anginal agent, endothelin receptor antagonist, phosphodiester-ase type 5 inhibitor, prostacyclin analog and soluble guanylate cyclase stimulator.

ACE indicates angiotensin converting enzyme; ARBs, angiotensin II receptor blockers; eFI, electronic frailty index; SMD, standardised mean difference.

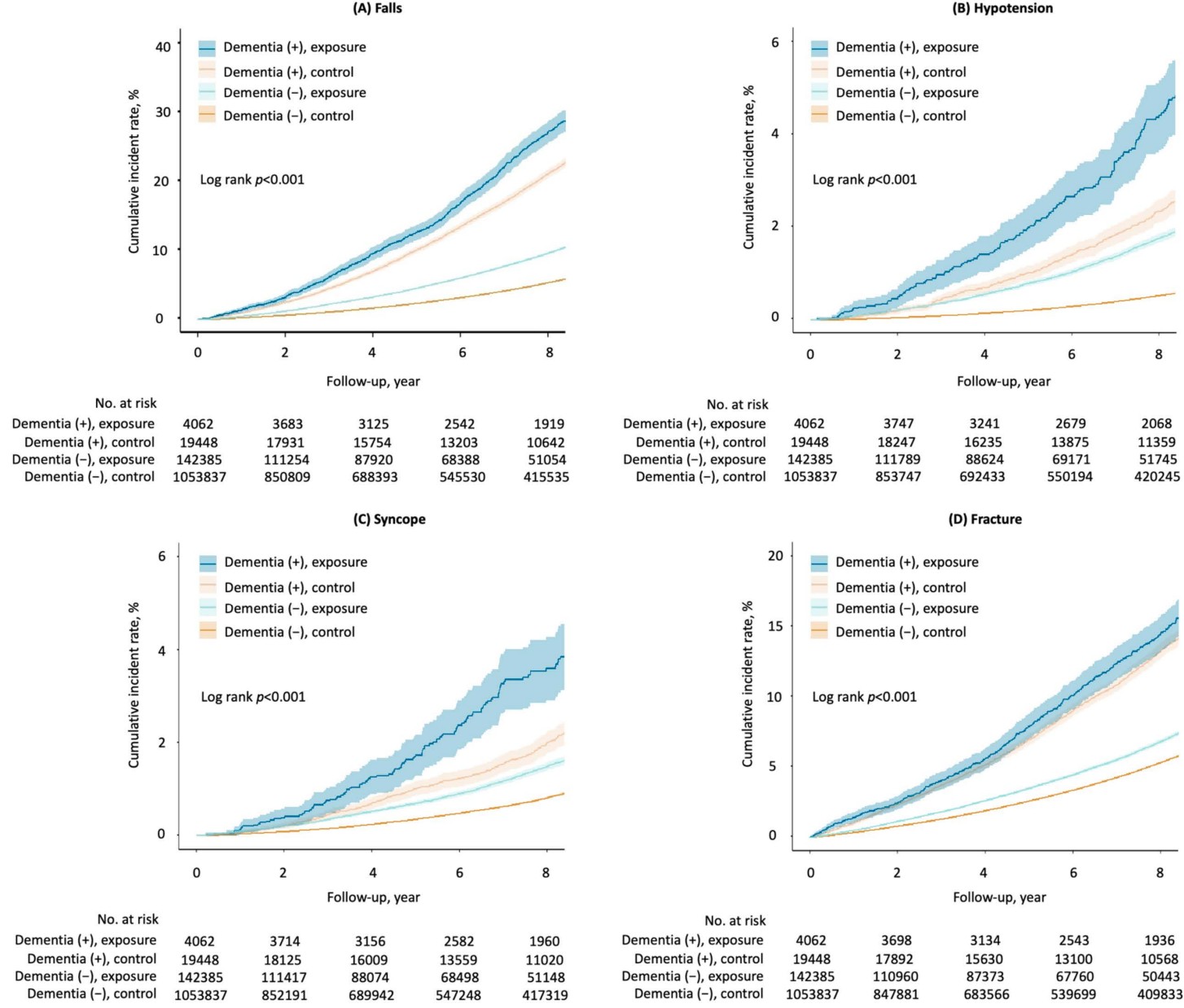

**Fig 1. Cumulative risk of serious adverse events by antihypertensive medication according to categories of dementia and drug exposure status.** Kaplan–Meier curves of the cumulative incidence of serious adverse events by the four categorical groups are shown. (**A**, **B**, **C**, and **D**) shows the cumulative incidence of falls, hypotension, syncope, and fracture, respectively. Each solid line indicates the cumulative incident rate for each serious adverse event, and its surrounding area indicates the 95% confidence interval. For all p-values reported in the figures, the statistical test used was the log-rank test.

## Sensitivity analyses

Of all study individuals, 1,107,593 (90.8%) started their follow-up periods before April 1, 2014, while 112,139 (9.2%) started them after that date. Among these, 23,206 (2.1%) and 304 (0.27%) were diagnosed with dementia, respectively.

Among those who started their follow-up periods after April 1, 2014, 77 (25.3%) with dementia and 1,647 (1.5%) without dementia experienced hospitalisation or died following a fall. Antihypertensive treatment was associated with an increased

**Table 2. The association between the risk of hospitalisation or death from falls with antihypertensive treatment and dementia status in imputation datasets.**

| | With dementia | | | | | Without dementia | | | | | |
| --- | --- | --- | --- | --- | --- | --- | --- | --- | --- | --- | --- |
| | Exposure group | | Control group | | | Exposure group | | Control group | | | P value for interaction |
| | Population | Event | Population | Event | Hazard ratio (95% CI) | Population | Event | Population | Event | Hazard ratio (95% CI) | |
| **Falls** (primary outcome) | | | | | | | | | | | |
| Propensity score adjustment | 4,062 | 1,533 | 19,448 | 6,781 | 1.15 (1.08, 1.22) | 142,385 | 14,684 | 1,053,837 | 73,101 | 1.07 (1.05, 1.10) | <0.001 |
| Multivariable adjustment | 4,062 | 1,533 | 19,448 | 6,781 | 1.12 (1.04, 1.19) | 142,385 | 14,684 | 1,053,837 | 73,101 | 1.10 (1.08, 1.12) | 0.034 |
| Propensity score matching | 3,795 | 1,442 | 3,795 | 1,396 | 1.12 (1.04, 1.22) | 131,199 | 13,383 | 131,199 | 13,576 | 1.06 (1.03, 1.08) | 0.043 |
| Propensity score IPTW | 4,062 | 1,533 | 19,448 | 6,781 | 1.18 (1.09, 1.28) | 142,385 | 14,684 | 1,053,837 | 73,101 | 1.21 (1.18, 1.25) | 0.599 |

Exposure group and control group indicates with antihypertensive prescription and without antihypertensive prescription, respectively. CI indicates confidence interval; IPTW, inverse probability of treatment weighting.

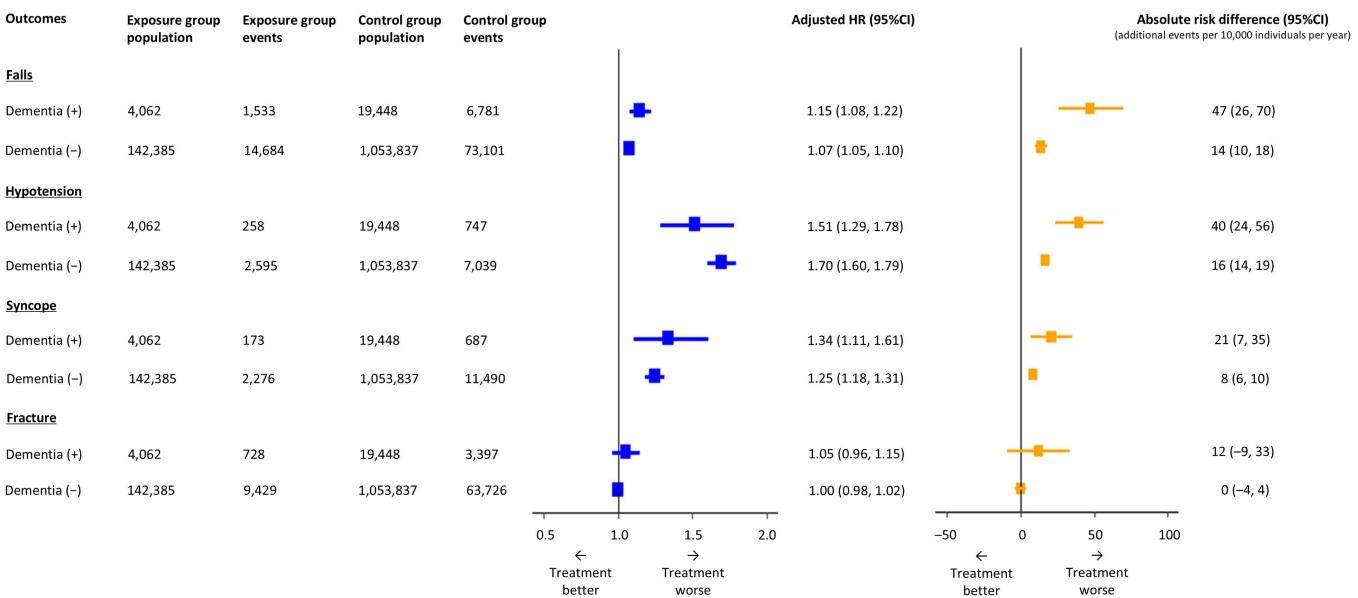

**Fig 2. Differences in the risk of serious adverse events associated with antihypertensive treatment by dementia status.** Models adjusted for propensity score. Absolute risk differences were described as additional events per 10,000 patients per year. Abbreviations: CI, confidence interval; HR, hazard ratio.

risk of hospitalisation or death from a fall in individuals without dementia (aHR 1.25, 95%CI 1.08, 1.46), but not in those with dementia (aHR 1.15, 95%CI 0.59, 2.27) (S9 Table).

## Discussion

In this nationwide electronic health record data-based observational study of 1.2 million previously untreated individuals, antihypertensive treatment was associated with an increased risk of hospitalisation or death from falls, hypotension, and

**Table 3. Hazard ratios of the initiation of antihypertensive medication drugs for each outcome in imputation datasets.**

| | With dementia | | | | | Without dementia | | | | |
|---|---|---|---|---|---|---|---|---|---|---|
| | Exposure group | | Control group | | | Exposure group | | Control group | | |
| | Population | Event | Population | Event | Hazard ratio (95% CI) | Population | Event | Population | Event | Hazard ratio (95% CI) |
| **Hypotension** | | | | | | | | | | |
| Propensity score adjustment | 4,062 | 258 | 19,448 | 747 | 1.51 (1.29, 1.78) | 142,385 | 2,595 | 1,053,837 | 7,039 | 1.70 (1.60, 1.79) |
| Multivariable adjustment | 4,062 | 258 | 19,448 | 747 | 1.48 (1.26, 1.75) | 142,385 | 2,595 | 1,053,837 | 7,039 | 1.63 (1.54, 1.72) |
| Propensity score matching | 3,800 | 230 | 3,800 | 169 | 1.52 (1.21, 1.90) | 131,186 | 2,309 | 131,186 | 1,487 | 1.60 (1.48, 1.72) |
| IPTW | 4,062 | 258 | 19,448 | 747 | 1.55 (1.29, 1.86) | 142,385 | 2,595 | 1,053,837 | 7,039 | 1.83 (1.70, 1.96) |
| **Syncope** | | | | | | | | | | |
| Propensity score adjustment | 4,062 | 173 | 19,448 | 687 | 1.34 (1.11, 1.61) | 142,385 | 2,276 | 1,053,837 | 11,490 | 1.25 (1.18, 1.31) |
| Multivariable adjustment | 4,062 | 173 | 19,448 | 687 | 1.27 (1.05, 1.54) | 142,385 | 2,276 | 1,053,837 | 11,490 | 1.25 (1.18, 1.31) |
| Propensity score matching | 3,792 | 166 | 3,792 | 129 | 1.36 (1.06, 1.75) | 131,248 | 2,081 | 131,248 | 1,839 | 1.21 (1.12, 1.31) |
| IPTW | 4,062 | 173 | 19,448 | 687 | 1.32 (1.06, 1.64) | 142,385 | 2,276 | 1,053,837 | 11,490 | 1.36 (1.26, 1.46) |
| **Fracture** | | | | | | | | | | |
| Propensity score adjustment | 4,062 | 728 | 19,448 | 3,397 | 1.05 (0.96, 1.15) | 142,385 | 9,429 | 1,053,837 | 63,726 | 1.00 (0.98, 1.02) |
| Multivariable adjustment | 4,062 | 728 | 19,448 | 3,397 | 1.04 (0.95, 1.13) | 142,385 | 9,429 | 1,053,837 | 63,726 | 1.02 (0.99, 1.04) |
| Propensity score matching | 3,788 | 676 | 3,788 | 686 | 1.05 (0.93, 1.18) | 131,335 | 8,672 | 131,335 | 9,252 | 0.99 (0.96, 1.03) |
| IPTW | 4,062 | 728 | 19,448 | 3,397 | 1.03 (0.92, 1.15) | 142,385 | 9,429 | 1,053,837 | 63,726 | 1.08 (1.05, 1.12) |

Exposure group and control group indicates with antihypertensive prescription and without antihypertensive prescription, respectively. CI indicates confidence interval; IPTW, inverse probability treatment weighting.

syncope in both individuals with and without dementia, with comparable relative risks (RRs) observed between the two groups. However, the absolute risk of harm with antihypertensive treatment was significantly higher in individuals with dementia than in those without dementia. These findings indicate that the absolute risk of serious adverse events with antihypertensive drugs differs by dementia status, and that careful consideration of the priorities of individuals with dementia is necessary when starting new antihypertensive medications in this group.

There is inconsistent evidence regarding the association of serious adverse events with antihypertensive therapy in dementia. In an observational study of 160 individuals living with dementia in group dwellings (mean age 84±7 years, 80% female), antihypertensive medication use was not a significant risk factor for falls [37]. On the other hand, in the Hypertension in Dementia study of 180 individuals with diagnosed hypertension and dementia (mean age 82±6 years, 70% female, 87% living their own home), a total of 214 falls (a rate of 2,760 falls per 1,000 patient-years) were observed during the 6-month follow-up, in individuals prescribed a median of one antihypertensive medication [38]. These data could not determine the association between serious adverse events by antihypertensive therapy and dementia because of very limited sample size, and lack of a non-treated control group. In contrast, our analyses examined a large population from general practice.

There are some possible mechanisms by which antihypertensive treatment could have adverse effects in individuals with dementia, such as orthostatic BP changes [39], accumulation of anticholinergic effects from antihypertensive drugs [40], drug-induced delirium [41], and drug interaction [42]. Frailty, social, environmental, and cognitive factors could also explain the higher absolute risk of falls in individuals with dementia [43–45]. Furthermore, our findings suggest that dementia itself may amplify the risk of serious adverse events associated with antihypertensive treatment. In our cohort, individuals with dementia had a markedly higher absolute risk of falls compared to those without dementia, despite similar RRs, supporting the notion that dementia-related vulnerability may enhance the harms of treatment.

**Table 4. Hazard ratios by the number of antihypertensive medications use for each outcome in imputation datasets.**

| Number of antihypertensive drugs | With dementia | | | | Without dementia | | | |
|---|---|---|---|---|---|---|---|---|
| | Population | Event | Hazard ratio (95% CI) | P value for trend | Population | Event | Hazard ratio (95% CI) | P value for trend |
| **Falls** (primary outcome) | | | | | | | | |
| 0 | 19,448 | 6,781 | – (reference) | 0.005 | 1,053,837 | 73,101 | – (reference) | 0.050 |
| 1 | 2,397 | 906 | 1.16 (1.08, 1.25) | | 86,394 | 8,733 | 1.09 (1.07, 1.12) | |
| 2 | 1,147 | 417 | 1.11 (1.00, 1.23) | | 38,826 | 4,035 | 1.02 (0.99, 1.06) | |
| ≥3 | 518 | 210 | 1.26 (1.09, 1.45) | | 17,165 | 1,916 | 1.07 (1.02, 1.13) | |
| **Hypotension** | | | | | | | | |
| 0 | 19,448 | 747 | – (reference) | <0.001 | 1,053,837 | 7,039 | – (reference) | <0.001 |
| 1 | 2,397 | 144 | 1.47 (1.22, 1.78) | | 86,394 | 1,351 | 1.58 (1.49, 1.69) | |
| 2 | 1,147 | 73 | 1.50 (1.16, 1.94) | | 38,826 | 795 | 1.78 (1.64, 1.93) | |
| ≥3 | 518 | 41 | 1.82 (1.30, 2.54) | | 17,165 | 449 | 2.18 (1.97, 2.42) | |
| **Syncope** | | | | | | | | |
| 0 | 19,448 | 687 | – (reference) | 0.078 | 1,053,837 | 11,490 | – (reference) | <0.001 |
| 1 | 2,397 | 93 | 1.22 (0.97, 1.53) | | 86,394 | 1,313 | 1.21 (1.14, 1.29) | |
| 2 | 1,147 | 58 | 1.60 (1.21, 2.13) | | 38,826 | 665 | 1.32 (1.21, 1.44) | |
| ≥3 | 518 | 22 | 1.38 (0.89, 2.14) | | 17,165 | 298 | 1.35 (1.20, 1.52) | |
| **Fracture** | | | | | | | | |
| 0 | 19,448 | 3,397 | – (reference) | 0.605 | 1,053,837 | 63,726 | – (reference) | 0.250 |
| 1 | 2,397 | 434 | 1.07 (0.97, 1.19) | | 86,394 | 5,748 | 1.02 (0.99, 1.05) | |
| 2 | 1,147 | 210 | 1.08 (0.93, 1.25) | | 38,826 | 2,481 | 0.96 (0.92, 1.00) | |
| ≥3 | 518 | 84 | 0.94 (0.75, 1.18) | | 17,165 | 1,200 | 1.06 (1.00, 1.13) | |

The results of Cox regression analyses adjusted for propensity score were shown. The number of individuals with dementia taking more than three antihypertensive medications were as follows: 3 drugs, $n=400$; 4 drugs, $n=97$; 5 drugs, $n=17$; 6 drugs, $n=4$; 7 drugs, $n=0$. The number of individuals without dementia taking more than three antihypertensive medications were as follows: 3 drugs, $n=13,044$; 4 drugs, $n=3,365$; 5 drugs, $n=660$; 6 drugs, $n=92$; 7 drugs, $n=4$. The average numbers of antihypertensive medications in individuals with and without dementia were $0.27\pm0.69$ and $0.18\pm0.57$, respectively.

There are very limited data regarding the benefits and harms of antihypertensive medications use in individuals with dementia. In a double-blind randomised controlled trial, the calcium channel blocker, nilvadipine, increased cerebral blood flow in individuals with mild-to-moderate Alzheimer's disease [17]. Additionally, another open-label randomised controlled trial demonstrated that in older hypertensive individuals with Alzheimer's disease, the angiotensin II receptor blockers, telmisartan, increased regional cerebral blood flow, including the superior parietal lobe which was the most severely affected region in Alzheimer's disease, compared to the calcium channel blocker, amlodipine [18]. One randomised controlled trial investigated the effects of the calcium channel blocker, nimodipine, in individuals with subcortical vascular dementia [46]. This study showed placebo was associated with increased risks of cerebrovascular disease (RR 2.48, 95%CI 1.23, 4.98) and CVD (RR 2.26, 95% CI 1.11, 4.60) compared with nimodipine treatment group. However, the favourable effects of nimodipine did not depend on the BP-lowering effect of nimodipine, and the original aim of this study was not to test the efficacy of nimodipine in the secondary prevention of vascular diseases. Furthermore, a recent randomised controlled trial among long-term care residents with moderate-to-severe dementia suggested that discontinuation of antihypertensive treatment tended to increase the risk of serious adverse events, although this trial was not powered on clinical outcome events and had to be stopped early, precluding any definitive conclusions from being drawn [47]. The potential benefit of BP-lowering effects may be modified by the degree of cognitive decline. In the Longitudinal Ageing Study Amsterdam, lower diastolic BP was associated with higher all-cause mortality risk in individuals with cognitive dysfunction [48]. In a

cohort study of individuals with diagnosed mild cognitive impairment or dementia, lower daytime systolic BP was associated with greater progression of cognitive decline among individuals with antihypertensive drug therapy [19].

While our previous observational study, which included the same individuals as the present analysis, demonstrated a similar association between antihypertensive treatment and serious adverse events [9], a meta-analysis of randomised controlled trials reported no significant association between antihypertensive treatment and fall risk [49]. This discrepancy may be partly explained by differences in study population, particularly the exclusion of frailer individuals and people with dementia from many clinical trials, in contrast to our real-world cohort. It may also reflect the influence of residual confounding, such as confounding by indication, which is more likely to affect observational data [50].

Recent guidelines for treatment of hypertension recommend an individualised approach for people with dementia [10–12]. Some antihypertensive drugs for individuals with dementia may have beneficial effects for cerebral blood flow [17,18], but there is no definitive evidence to date for benefit from antihypertensive therapy on CVD outcomes in individuals with dementia [51,52], because most trial participants were relatively fit, people with dementia were excluded by design [4,5,53]. Our findings suggest that, in line with previous literature [54–57], the cumulative event rates of falls, hypotension, syncope, and fractures and absolute risk differences of them were higher in individuals with dementia than in those without. Various factors, such as advanced age [58,59], sleep-related disorders [60,61], depression [62], autonomic dysfunction [63], socioeconomic factors [43], and quality of care could affect the association between the increase of harmful events and dementia [64]. Our current findings extend these previous findings and suggest that new prescriptions of antihypertensive drugs might increase the risk of harmful events in individuals with dementia. Furthermore, some individuals with dementia may have different health goals when considering preventive therapy, for example, maximising quality of life and avoiding hospitalisation due to serious adverse events may be more important than preventing fatal CVD events [65]. Previous work found that clinicians encountered numerous challenges when optimising prescribing for individuals with dementia, including decisions about stopping medication [66]. Our findings can inform such clinical decision-making and clinicians managing hypertension in individuals with dementia should consider the balance of risks and benefits. Clinicians should pursue patient-centred care, where the patient's goals and wishes are prioritised in agreeing the treatment.

Formal interaction tests between dementia status and antihypertensive treatment were statistically significant in three of the four analytical models, suggesting a modest but statistically detectable difference in RR by dementia status. However, the magnitude of effect modification was small, and given the large sample size, these findings should be interpreted cautiously as they may have limited clinical relevance.

Strengths of this study include the large general practice population-based sample and the examination of how the association between antihypertensive therapy and serious adverse events differ by dementia status. Despite the strengths of the study, it should be interpreted within the context of its potential limitations. First, while antihypertensive treatment is known to reduce the risk of cardiovascular mortality, our study was not designed to evaluate the potential cardiovascular benefits of antihypertensive treatment in individuals with dementia. As such, our findings should not be interpreted as a comprehensive risk-benefit assessment of antihypertensive therapy in this population. Second, dementia might have been underdiagnosed in this study population, which might have affected the study results. The association between serious adverse events risk with antihypertensive medication and dementia status diagnosed after April 2014 could not be reliably assessed in the present study due to very limited number of dementia cases after this date. Furthermore, although CPRD GOLD is a widely validated and high-quality UK primary care database, the accuracy and completeness of diagnostic coding for dementia and related comorbidities may be imperfect. Previous validation studies have demonstrated acceptable levels of diagnostic accuracy for these conditions; however, under-ascertainment or misclassification remains possible and may have influenced our study findings [26,67,68]. Third, the study included individuals with different types of dementia, based on coded diagnoses in the electronic health record, rather than careful cognitive testing. This heterogeneity of dementia diagnosis could also affect the results, as previous study showed that vascular dementia, mixed dementia, and dementia in other diseases were associated with increased risk of falls compared to Alzheimer's disease [43]. In addition,

this study did not include individuals with mild cognitive impairment who may also be at risk of serious adverse events. Fourth, we used an "intention-to-treat approach" and did not account for individuals who developed new dementia during the observation period or who initiated antihypertensive treatment in the control group (28.9% during follow-up). However, the median treatment duration was 6 years in the exposure group versus 0 years in the control group, suggesting that our effect estimates likely reflect conservative estimates of the true treatment-associated risk. More complex analyses using time-varying covariates could theoretically refine exposure classification [69]; however, integrating time-varying exposure into an analysis already involving multiple imputation and propensity score adjustment across a large, real-world dataset poses considerable methodological challenges, which are not well understood in the literature. Moreover, this approach likely yields conservative risk estimates, as control individuals who initiated treatment during follow-up were not reclassified, which may have attenuated observed treatment effects. In addition, time-varying exposure in observational data is highly susceptible to confounding by indication, as treatment initiation during follow-up may reflect worsening clinical status, thereby introducing bias that is difficult to fully address. In addition, when exposure status changes over time and may itself be influenced by prior outcomes, fixed-exposure models may be particularly vulnerable to structural confounding. While causal methods such as G-methods have been developed to address this issue [70], their implementation would require further assumptions and complexity that go beyond the scope of the current study. Furthermore, by treating exposure as fixed over time, our analysis may be subject to additional unmeasured confounding arising from changes in treatment status that are not captured in our model. We also note that we did not evaluate subsequent treatment changes, including discontinuation or dose modification of antihypertensive drugs, during the follow-up period in the exposure group. Fifth, we could not assess adherence and persistence of antihypertensive medications during the observation period. Some people with dementia have poor adherence to antihypertensive medication [71], while others might have better adherence than those without dementia due to support from the community and carers [72], which might lead to an underestimation or overestimation of the potential association between antihypertensive treatment and serious adverse events. In addition, although the dosage of antihypertensive medications is associated with the risk of serious adverse events [73], we could not assess this relationship in our study. Sixth, our findings may not be generalisable to individuals with advanced dementia, given evidence from the previous study linking sedentary behaviour to decreased cognitive function [74]. Seventh, although our propensity score analyses were successful in balancing the groups based on known confounding variables, we cannot rule out the presence of unmeasured confounding [49,75]. Nevertheless, the relative comparison between individuals with and without dementia remains valid, since the treatment effects in both individuals with and without dementia would be subject to the same potential confounders. Eighth, while we took steps to minimise the impact of missing data through appropriate imputation methods, a high proportion of individuals required imputation for a limited number of variables. Our analysis showed only a slight difference between the imputation datasets and the complete-case dataset, indicating that the imputation might introduce minimal bias. However, the findings should still be interpreted with caution due to the presence of missing variables. Ninth, while the CPRD Gold database is representative of the UK population in terms of ethnicity [26], potential underreporting of dementia among minority ethnic groups may result in an overrepresentation of White individuals in our dataset. Tenth, we did not use a Fine-Gray competing risks model because our primary aim was to estimate cause-specific hazards rather than cumulative incidence, and individuals who died from unrelated causes were censored accordingly. Moreover, previous analyses using the same dataset have shown that results from Cox and Fine-Gray models were largely consistent [9], supporting the robustness of our approach. Finally, our findings may not be generalisable to different populations or ethnic groups.

Taken together, in previously untreated individuals with elevated systolic BP, this study demonstrated the increased absolute risk of serious adverse events associated with antihypertensive treatment in individuals with dementia compared to those without dementia. When prescribing antihypertensive medication for individuals with dementia, careful consideration of potential adverse events is needed. Clinicians, patients, and their carers should carefully consider the balance between benefits and harms of antihypertensive treatment for individuals with dementia.

## Supporting information

**S1 Protocol. ISAC Protocol.**
(DOCX)

**S1 Table. The RECORD statement.**
(DOCX)

**S2 Table. The CPRD GOLD medical codes used to define dementia.** CPRD indicates Clinical Practice Research Datalink.
(DOCX)

**S3 Table. Drug types and corresponding British National Formulary header included in the analysis.**
(DOCX)

**S4 Table. Propensity score model in imputation datasets.** BP indicates blood pressure; CI, confidence interval; DBP, diastolic blood pressure; FI, frailty index; HDL, high-density lipoprotein; IMD, indices of multiple deprivation; SBP, systolic blood pressure.
(DOCX)

**S5 Table. Propensity score model in the complete-case dataset.** BP indicates blood pressure; CI, confidence interval; DBP, diastolic blood pressure; FI, frailty index; HDL, high-density lipoprotein; IMD, indices of multiple deprivation; SBP, systolic blood pressure.
(DOCX)

**S6 Table. Baseline characteristics of the study population before or after propensity score matching in the complete-case dataset.** Data are mean ± SD or number (percentage). A standardised mean difference (SMD) of <0.1 suggests adequate variable balance after propensity score matching. *High deprivation indicates indices of multiple deprivation score of 5 (most deprived). †Thiazides includes thiazide-like diuretics. ‡Other antihypertensives includes centrally acting antihypertensives, direct renin inhibitors and vasodilators. ACE indicates angiotensin converting enzyme; ARBs, angiotensin II receptor blockers; eFI, electronic frailty index; SMD, standardised mean difference.
(DOCX)

**S7 Table. Hazard ratios of the initiation of antihypertensive medication drugs for each outcome in the complete-case dataset.** Exposure group and control group indicates with antihypertensive prescription and without antihypertensive prescription, respectively. CI indicates confidence interval; IPTW, inverse probability of treatment-weighted.
(DOCX)

**S8 Table. Hazard ratios by the number of antihypertensive medications use for each outcome in the complete-case dataset.** The results of Cox regression analyses adjusted for propensity score were shown. The number of individuals with dementia taking more than three antihypertensive medications were as follows: 3 drugs, $n = 150$; 4 drugs, $n = 43$; 5 drugs, $n = 9$; 6 drugs, $n = 3$; 7 drugs, $n = 0$. The number of individuals without dementia taking more than three antihypertensive medications were as follows: 3 drugs, $n = 4,001$; 4 drugs, $n = 1,133$; 5 drugs, $n = 257$; 6 drugs, $n = 42$; 7 drugs, $n = 2$.
(DOCX)

**S9 Table. Hazard ratios of the initiation of antihypertensive medication drugs for a fall by the start date of the follow-up period.** Exposure group and control group indicates with antihypertensive prescription and without antihypertensive prescription, respectively. CI indicates confidence interval; IPTW, inverse probability treatment weighting.
(DOCX)

**S1 Fig. Definition of time periods used to define the cohort and follow-up periods.** Individuals were eligible for cohort entry if they met the following criteria: (1) aged ≥40 years old; (2) with qualifying first systolic BP levels of between 130 − 179 mmHg prior to the exposure period; (3) not having received any antihypertensives prior to the study start date; and (4) who were registered between 1st January 1998 and 31st December 2018 in CPRD GOLD.
(DOCX)

**S2 Fig. Flow diagram showing selection of patient records for inclusion in the study.**
(DOCX)

**S3 Fig. Distribution of propensity scores by treatment status in individuals with and without dementia. (A)** the distribution of propensity scores among individuals with dementia, and **(B)** the distribution among those without dementia.
(DOCX)

**S4 Fig. Cumulative risk of serious adverse events by antihypertensive medication according to categories of dementia and drug exposure status in the complete-case dataset.** Kaplan–Meier curves of the cumulative incidence of serious adverse events by the four categorical groups in complete case are shown. S3A, S3B, S3C, and S3D Fig shows the cumulative incidence of falls, hypotension, syncope, and fracture, respectively. Each solid line indicates cumulative incident rate for each serious adverse event and its around area indicates the 95% confidence interval.
(DOCX)

**S5 Fig. Differences in the risk of serious adverse events associated with antihypertensive treatment by dementia status in the complete-case dataset.** Models adjusted for propensity score using complete cases only. Absolute risk differences were described as additional events per 10,000 patients per year. CI indicates confidence interval; and HR, hazard ratio.
(DOCX)

## Acknowledgments

We would like to acknowledge all investigators involved in the study. We thank Richard Stevens, medical statistician at Nuffield Department of Primary Care Health Sciences, University of Oxford, for giving us important contributions to creating the programme of STRATIFY project. This work uses data provided by patients and collected by the NHS as part of their care and support. We are very grateful to all those patients who permit their anonymised routine NHS data to be used for this approved research. The views expressed are those of the authors and not necessarily those of the NIHR or the Department of Health and Social Care.

## Author contributions

**Conceptualisation:** Takeshi Fujiwara, James P. Sheppard.

**Data curation:** Takeshi Fujiwara, Constantinos Koshiaris, Ting Cai, Ariel Wang, James P. Sheppard.

**Formal analysis:** Takeshi Fujiwara.

**Investigation:** Takeshi Fujiwara.

**Methodology:** Takeshi Fujiwara, Constantinos Koshiaris, Ting Cai, Ariel Wang, Joseph Lee, James P. Sheppard.

**Project administration:** Takeshi Fujiwara, James P. Sheppard.

**Supervision:** Constantinos Koshiaris, Ting Cai, Ariel Wang, Joseph Lee, Sarah Lay-Flurrie, Amitava Banerjee, Andrew Clegg, Rupert A. Payne, Subhashisa Swain, Margaret Ogden, Satoshi Hoshide, Kazuomi Kario, F. D. Richard Hobbs, Richard J. McManus, James P. Sheppard.

**Writing – original draft:** Takeshi Fujiwara.

**Writing – review & editing:** Constantinos Koshiaris, Ting Cai, Ariel Wang, Joseph Lee, Sarah Lay-Flurrie, Amitava Banerjee, Andrew Clegg, Rupert A. Payne, Subhashisa Swain, Margaret Ogden, Satoshi Hoshide, Kazuomi Kario, F. D. Richard Hobbs, Richard J. McManus, James P. Sheppard.

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
