## [Editor Report · Decision Letter 0]

8 Jan 2025

Dear Dr Sheppard, 

Thank you for submitting your manuscript entitled "Associations between falls and other serious adverse events and antihypertensive medication in individuals with dementia: An observational cohort study" for consideration by PLOS Medicine.

Your manuscript has now been evaluated by the PLOS Medicine editorial staff as well as by an academic editor with relevant expertise and I am writing to let you know that we would like to send your submission out for external peer review.

Please re-submit your manuscript within two working days, i.e. by Jan 10 2025 11:59PM.

Kind regards,

Suzanne

Suzanne De Bruijn, PhD

Senior Editor

PLOS Medicine

---

## [Decision Letter · Decision Letter 1]

28 Apr 2025

Dear Dr Sheppard,

Many thanks for submitting your manuscript "Associations between falls and other serious adverse events and antihypertensive medication in individuals with dementia: An observational cohort study" (PMEDICINE-D-25-00036R1) to PLOS Medicine. Please accept my apology for the unusual delay in providing you with a decision. The paper has now been reviewed by subject experts and a statistician; their comments are included below and can also be accessed here: [LINK]

As you will see, the reviewers find the question interesting but have some serious concerns regarding the analysis as well. After discussing the paper with the editorial team and an academic editor with relevant expertise, I'm pleased to invite you to revise the paper in response to the reviewers' comments. We would need to see all the reviewer concerns addressed, but specifically the issue the time-varying exposure. I have included some additional comments from our statistical editor below. However, after discussion with the Academic Editor, we think the concerns regarding only assessing 'risks' and not 'benefits' can be addressed in the discussion, rather than with additional analyses. We plan to send the revised paper to some or all of the original reviewers, and we cannot provide any guarantees at this stage regarding publication.

We ask that you submit your revision by May 19 2025 11:59PM. However, if this deadline is not feasible, please contact me by email, and we can discuss a suitable alternative.

Don't hesitate to contact me directly with any questions (sbruijn@plos.org). 

Best regards, 

Suzanne 

Suzanne De Bruijn, PhD 

Associate Editor

PLOS Medicine

sbruijn@plos.org

Comments from the academic editor:

The reviewers have raised some serious concerns that should be addressed. However, the issue of not examining 'benefits' is less concerning, as this is covered in the literature. Although this issue therefore does not need to be addressed in the analysis, they should include this in the discussion. 

Comments from the reviewers: 

Reviewer #1: The data are what they are in an observational study, thank you for the revision.

Reviewer #2: The present one is a retrospective longitudinal study comparing hospitalization and/or death due to fall, hypotension, syncope and fracture between subjects prescribed and not prescribed with antihypertensive drugs (AHD). The stated primary object of the study Is the comparison of AHD-associated risk between subjects diagnosed and not diagnosed with dementia. The Authors conclude that AHD-associated relative risk of fall, hypotension and syncope (but not fracture) is similarly increased for subjects with and without dementia, why absolute risk increase is greater in dementia subgroup, which therefore would need a more cautious risk-to-benefit assessment. 

The present data offer new information regarding a clinically relevant issue. Yet I have some observations.

Authors state that their primary aim is to examine the association between AHD and serious adverse events in individuals with dementia compared to those without dementia. Yet in statistical analysis section the association is assessed within each group, but no direct measure of comparison is indicated as outcome. Please improve data analysis on this point. 

In particular, regarding relative risk of adverse events, Authors conclude that it was "comparable between the two groups", but report among results, and in supplementary table S7, that significant interactions were observed in the association of primary outcome with antihypertensive treatment and dementia status. In Supplementary Table S7 actually it is reported that, in 3 out of the 4 parallel statistical methods they used, the interaction was significant. Therefore I understand that relative risk was actually greater, although slightly, among subjects with dementia. I feel that Table S7 should be reported as main result, and Authors should discuss why inverse probability weighting analysis does not confirm the increased relative risk in subjects with vs. without dementia.

Regarding absolute risk increase, this appears to be clearly greater among dementia subjects, but no statistical test is reported to confirm this difference, which is presented as the main study result. Formal statistical testing should be reported for this primary outcome. Moreover, at lines 392-396 Authors discuss specific mechanisms explaining why "antihypertensive treatment could have adverse effects in individuals with dementia". Yet, according to the data, the incidence of the primary outcome was clearly greater in dementia subgroup irrespectively of AHD status (5x), and I feel that the Authors should state that this probably explains largely by himself the greater absolute risk observed in among subjects treated with AHD in the dementia vs. non-dementia subgroup (about 3.5x).

Authors extensively discuss study limitations, but should better address the risk of unmeasured confounding, as the sample of subjects treated with AHD show a huge clinical difference, as expected, in comparison with the non-treated. In particular

they are to be congratulated as they used parallel analyses, including propensity score adjusting and matching, which confirm that the association of AHD with hospitalization/death for fall, hypotension and syncope is independent of the greater clinical complexity of treated subjects. I feel that the use of propensity score should be cited in the Abstract too;

they state that any potential effect of unmeasured confounding would be limited, as "the overall effects of antihypertensive treatment on serious adverse events in this population was similar to findings from meta-analyses of randomised controlled trials". Yet in this metaanalysis (that Authors should include among references: Albasri A, Hattle M, Koshiaris C, Dunnigan A, Paxton B, Fox S E et al. Association between antihypertensive treatment and adverse events: systematic review and meta-analysis BMJ 2021; 372: n189) the risk of falls was similar between treated and non-treated subjects. This suggests that at least part of the increased risk of death/hospital admission due to fall associate with AHD (irrespectively of dementia) in this sample is probably the expression of a "confounding by indication" linked to a greater vulnerability not fully captured by measured covariates;

therefore, they should moderate their conclusions, suggesting that, due to observational design of the study, part of the increased risk might be explained by unmeasured confounding

Authors state that they did not use a competing risks model to account for competing mortality risks, as according to similar analyses they performed in the same sample (ref 6), "this is unlikely to have resulted in substantially different results". This may well be the case. Yet, as one the main aims of AHD is to reduce vascular mortality, I feel that at least an adjusted comparison of total (and possibly vascular) mortality between treated and non-treated subjects should be provided to give the reader a clearer picture of vulnerability of the two sample and of possible trade-off between risk and benefits of AHD 

Authors should provide, possible in a supplementary table, separate figures of deaths and hospitalizations associated with different outcomes within the 2 subgroups, to give the reader a greater awareness of the events that are driving the observed associations and to better estimate trade-offs of treatment (I expect hospitalizations to have a greater weight than deaths)

Minor observation:

please rephrase at line 71 and following: "the absolute INCREASE OF FALL RISK ASSOCIATED with antihypertensive treatment per 10,000 individuals per year was

significantly higher…".

Reviewer #3: Thanks for the opportunity to read your manuscript. My role is statistical reviewer, so I have focused on the design, data, and analysis that are presented. I have put general comments first, followed by questions relevant to a specific section of the manuscript (with a page/line reference). 

This manuscript presents a retrospective cohort study examining differences in risk of hospitalisation for falls (and hospitalisation for hypotension, syncope, and fracture) associated with antihypertensive medicines between people with and without dementia in the United Kingdom. Data is from UK CPRD, which includes linkage between primary care data, hospitalisations, deaths, and prescribing data. The codes used for this project are all provided on a GitHub repository. Dementia was defined from clinical codes, exposure to anti-hypertensive agents was from prescribing data. People without BP data, or extreme hypertension without a record or antihypertensive prescribing were excluded. Outcomes were defined from mortality records and hospitalisations, the primary outcome was falls, and secondary outcomes of hypotension, syncope, or fracture were included. There was a relatively high amount of missing data (particularly cholesterol measurements), multiple imputation (MICE with 10 datasets) was used for the main analyses. Propensity score matching (1:1) was used, with the PS based on the available covariates. The main analysis on the matched data were Cox models, with 4 approaches to adjusting for confounders considered (with the expectation that they would have similar results). Several subgroup and sensitivity analyses were considered, including by number of antihypertensive agents, and study period (because the coding for dementia may have changed over time). 

Matching was effective, and there was evidence that risk of falls was higher in those prescribed antihypertensive medications. The figures and tables are good quality, results were consistent across different approaches to account for confounding. 

Were the analyses undertaken here all as specified in the study protocol? I particularly wanted to check that the estimate of absolute risk was pre-planned. 

How was death from other causes dealt with? Was this with censoring of the study period? Is this likely to represent a competing risk to the main outcomes in this study? 

P12, L172. Was a fixed lookback period for BP used, or did this vary according to what data was available?

To clarify, was the first hypertensive episode in each person's data used as the inception point/baseline for the study? How were people who stopped taking antihypertensives or changed dose dealt with? 

P12, L175. Were these two exclusions made because these criteria indicated that these people had incomplete or poor-quality data?

P13, L183. Was dementia exposure defined just based on status at baseline? How were people who subsequently developed dementia dealt with in the analysis?

P14, L213. How was the number of imputations decided? Was this in line with the estimates of fraction of missing information from the main analyses? 

P15, L231. Why was a single-imputation strategy used for QRisk2, instead of including this in the MI model?

P16, L244. Was timing of start of baseline used in the matching process at all?

Was the propensity score estimated separately by dementia status? 

P17, L256. To clarify, was separate Cox model run for each level of dementia status, or just one model with a interaction between dementia status and antihypertensive exposure?

P19, L302. Was there evidence of common support with respect to the propensity score? 

Table 3. How was the p-value for 'trend' generated? Is this an overall test of heterogeneity, or from including the number of drugs as a continuous variable? 

Reviewer #4: Thank you for the opportunity to review this paper. The authors compared the risk of hospitalisation/death from falls between new users versus non-users of antihypertensive medications, over 10 years of follow-up, stratified by dementia status, using routine healthcare data from 1.2 million individuals aged 40 or older. Secondary outcomes were first hospitalisation/death from hypotension, syncope and fracture. Confounding was extensively controlled for using propensity score matching, additional adjustment, and inverse probability weighting. The authors found that the main outcome risk was higher in antihypertensive users than in non-users with dementia (7%) and without (15%). In terms of absolute risks, the number needed to harm for falls was substantially lower in those with dementia (213/year) versus those without (714/year). Secondary outcomes and sensitivity analyses showed similar results. The authors conclude that clinicians, patients and caregivers should consider these risks before starting antihypertensive treatment, especially in those with dementia.

The research question is interesting, and there is a severe dearth of knowledge on the use of antihypertensive treatment in older people with dementia. However, the current study only addresses on part of the question, the potential harms, without analysing the potential benefits (reduced cardiovascular disease/dementia/mortality rates). Thereby, it provides little information to help patients and clinicians evaluate the harm/benefit ratio. The narrative in the introduction and discussion is also leaning somewhat towards the harms of antihypertensives in older cognitively frail groups, without mentioning evidence on the benefits. The study conforms to the relevant RECORD guidelines for routinely collected data, and the methods are well described and thorough. However, I do have some important questions about the design that was chosen, and whether it is most appropriate for the current research question. Especially with exposure being based on a single time window, rather than using time-varying covariates, and some questions about the antihypertensive medications being considered likely not being used for primary hypertension.

Overall, although this is an interesting study with comprehensive analyses in a very large dataset, I think it currently does not provide sufficient new information to merit publication in your journal. It may after extensive revision, and evaluation of the authors' responses to my comments, but I think this would require a new review after resubmission.

Please find my point-by-point comments below

Introduction

"However, caution should be exercised, as older people are at increased risk of serious adverse events, especially in those with progression of frailty"

To be fair, other evidence does suggest that individuals with frailty may also be at increased benefit from intensive blood pressure lowering,(1,2) and guidelines are not absolutely sure how to approach it but generally agree that hypertension treatment is still recommended.(3) Therefore, I think this statement is somewhat more debated than is currently suggested.

1. https://www.ahajournals.org/doi/10.1161/CIRCULATIONAHA.123.064003

2. https://www.ahajournals.org/doi/10.1161/HYPERTENSIONAHA.124.24214

3. https://academic.oup.com/ageing/article/51/1/afab192/6410447?login=false

Methods:

"Eligible individuals were those: (1) aged ≥40 years old; (2) with qualifying first systolic BP levels of between 130−179 mmHg9 prior to the exposure period; (3) not having received any antihypertensives prior to the study start date; and (4) who were registered between 1st January 1998 and 31st December 2018 in CPRD GOLD."

How can you be sure that individuals had not received antihypertensive medications prior to the study start date?

Moreover, given the main study outcome of 'hospitalisation of death from a fall within 10 years of follow-up', what is the rationale for including individuals from age 40 or older? The event rates will be very low in the group <70, and since dementia generally occurs >65 years, this would create an artificial contrast when comparing the incidence rates in individuals with dementia and those without, which is mostly due to the age difference.

"Individuals characteristics were determined from information recorded at any point prior to the start of the follow-up period."

When did the 'follow-up' period start? Was the 'baseline' exposure period from 1998 to 1999? If individuals turned 40 after that, did they enter the database and then exit after 10 years? Because your mostly interested in events that generally occur in very old age, what would be the use of following someone from e.g. 1998 to 2008 from their 50th to their 60th, if they could also contribute to the analysis from their 60th to their 70th and be of much more interest to this research question?

"The exposure was prescription of any antihypertensive medication as defined in the British National Formulary (Supplementary Table S3). Individuals were allocated to the exposure group if they had been prescribed at least one antihypertensive medication during a 12-month period prior to the start date of the follow-up."

I think this is problematic for several reasons. 1.) The antihypertensive medications under study also include medications that are not primarily used for primary hypertension including loop diuretics and beta-blockers which are likely given (in combination with) heart failure, cardiac disease and/or arrythmia. This would cause confounding by indication in the analyses, because individuals with these types of conditions would have a worse prognosis a-priori. 2.) The propensity score matching/confounding adjustment might have captured some of this confounding by indication, but only in individuals whose diagnoses were accurately kept and dated in the database, so more information about the sensitivity of the database is essential. Furthermore, the propensity score adjusts for a combination of conditions, so these confounding by indication variables may be outweighed by other variables that are better predictors of the outcome but do not incur the same type of biases. It would be more prudent to exclude all individuals with relevant indications that may cause prescription of the included antihypertensive medication types of reasons other than hypertension altogether. 3.) Since antihypertensive treatment is generally given as long-term chronic treatment, considering individuals as exposed when having received 'at least one antihypertensive medication' during a 12-month period does not pertain to the individuals this study is interested in. Rather, those with single/sporadic treatment might have tried antihypertensive medication but did not tolerate it, felt that it was unnecessary, or received loop diuretics because of oedema. Furthermore, the 'unexposed' group may contain many individuals who received antihypertensive medication after the follow-up period. 

I wonder whether people with elevated blood pressure who do not receive subsequent antihypertensive treatment are comparable to those who do. The fact that they did not receive treatment may be itself an indication that the treating physician did not think they had hypertension (e.g. white coat hypertension), or that their blood pressure was normal a couple of visits later. Matching for blood pressure may not control for this type of indication bias, because the exact reason for refraining from antihypertensive treatment (as recommended by guidelines) is unknown.

Results:

"After matching, the SMD for all variables included in propensity score was reduced to below 0.1, indicating effective matching (Table 1 and 306 Supplementary Table S6)."

If you look at the individual (potentially) confounding characteristics, you see that many are significantly different, with the antihypertensive group consistently doing worse. Of course, these are small differences on a population level and only significant because of the large numbers but put together they may cause small differences in the outcome.

Discussion

I think the elephant in the room here is that the outcomes only include those expected to be worse with antihypertensive treatment (death/hospitalisation from falls, hypotension, syncope), while they should be weighed against the potential gains (lower rates of mortality, cardiovascular disease, dementia etc.). With the information gathered by this study, it is impossible for clinicians to weigh the harms against the benefits. The absolute risk of falls may be higher but this may be compensated by greater longevity and/or fewer cardiovascular events. Randomized evidence does suggest that ceasing antihypertensive medication in cognitively frail older people results in poorer outcomes,(1) suggesting there is a benefit of taking these medications. Therefore, the current study does not sufficiently substantiate its conclusion that clinicians, patients and carers should consider the risks of antihypertensive treatment beyond what is already known.(2) It merely provides one side of the argument. What is really needed is a careful consideration of the harms versus the benefits.(2)

1.) https://pubmed.ncbi.nlm.nih.gov/38970547/

2.) https://pmc.ncbi.nlm.nih.gov/articles/PMC10978346/

"This is the first study to demonstrate that the absolute risk of serious adverse

events with antihypertensive drugs differs by dementia status."

To warrant this conclusion, I think it would be far more appropriate to precisely match individuals with dementia to those without, and subsequently assess the risk of the primary outcome. Currently it is difficult to tell to what extent the differences between those with and without dementia are caused by the dementia group being very different form the other population in terms of age, comorbidity, and other relevant factors.

I think the discussion could do more to discuss the findings of antihypertensive trials in older cognitively frail populations (e.g. SPRINT, DANTON)(1,2) rather than narrowly focus on individuals with dementia, which is a very difficult group to include in hypertension RCTs.(3)

1. https://www.ahajournals.org/doi/10.1161/CIRCULATIONAHA.123.064003

2. https://pubmed.ncbi.nlm.nih.gov/38970547/

3. https://pmc.ncbi.nlm.nih.gov/articles/PMC10978346/

"In addition, this study did not include individuals with mild cognitive impairment who may also be at risk of serious adverse events."

This statement is not completely clear on whether these people excluded from the study or was mild cognitive impairment status simply not regarded in the analyses in any way.

"Third, we used an "intention-to-treat approach" and did not account for individuals who developed new dementia during the observation period or who started antihypertensive treatment in the control group."

I think this limitation merits a lot more discussion besides this simple sentence. Arguably, using a baseline exposure design with a 10-year follow-up is not the most appropriate approach for this research question. Why did you not use analysis with time varying covariates? This would avoid the problems with defining an exposure period, and because you would expect the relationship between antihypertensive medication use and falls to be more or less direct (as long as antihypertensives are taken, people are at increased risk), this would seem a more suitable approach at first glance. Explain why you chose the current method, and/or whether another approach like time-varying covariates would have been more appropriate.

"Ninth, we did not use a Fine-Gray competing risks model to account for competing risks, although previous analyses using the same data suggest this is unlikely to have resulted in substantially different results."

Rather than mentioning this as a limitation, please explain why not.

"Tenth, the absence of individual BP values in our dataset meant that we could not assess potential relationships between increased BP variability and the risk of falls."

If blood pressure values were missing from the dataset, how did you assess the systolic blood pressure levels between 130-179?

Would it have been interesting to assess whether the results differed in subgroups according to blood pressure preceding treatment initiation? (e.g. <150,150-160,>160 mmHg)

Additional comments from the statistical reviewer:

If the authors have a comparison of the difference between dementia/non-dementia of the association between AHD and SAEs they should do this directly. The most powerful way to accomplish this would be to include an interaction between AHD and dementia status, and test for evidence of difference in the beta coefficient.

The time-varying exposure is a good point – particularly given that many people will commence treatment with an AHD as they age. The gold-standard for dealing with this is the application of ‘G methods’. Using a fixed exposure covariate is a problem where someone’s outcome at an earlier time-point affects their exposure at a later time-point, e.g. someone experiences a fall and their AHD is changed to prevent hypotension and dizziness. These are fairly complicated methods – it would be ideal to use one of the ‘g methods’ to deal with AHD as a time-varying exposure, but at the very least the potential bias caused from a fixed exposure should be acknowledged as a limitation of study design.

I think the manuscript does a reasonable job of talking about the limits from unmeasured confounding in the discussion in terms of a fixed exposure. If the next version doesn’t include time-varying covariate with appropriate methods they should also acknowledge the issue of confounding that can occur when the exposure varies over time but is treated as fixed in the analysis.

---

* Please upload any figures associated with your paper as individual TIF or EPS files with 300dpi resolution at resubmission; please read our figure guidelines for more information on our requirements: http://journals.plos.org/plosmedicine/s/figures. While revising your submission, please upload your figure files to the PACE digital diagnostic tool, https://pacev2.apexcovantage.com/. PACE helps ensure that figures meet PLOS requirements. To use PACE, you must first register as a user. Then, login and navigate to the UPLOAD tab, where you will find detailed instructions on how to use the tool. If you encounter any issues or have any questions when using PACE, please email us at PLOSMedicine@plos.org.

FIGURES AND TABLES

SUPPLEMENTARY MATERIAL

REFERENCES

OBSERVATIONAL STUDIES

* Abstract: Please include the study design, population and setting, number of participants, years during which the study took place (enrollment and follow up), length of follow up, and main outcome measures.

* For all observational studies, in the manuscript text, please indicate: (1) the specific hypotheses you intended to test, (2) the analytical methods by which you planned to test them, (3) the analyses you actually performed, and (4) when reported analyses differ from those that were planned, transparent explanations for differences that affect the reliability of the study's results. If a reported analysis was performed based on an interesting but unanticipated pattern in the data, please be clear that the analysis was data driven. 

* Please state in the Methods section whether the study had a prospective protocol or analysis plan. If a prospective analysis plan (from your funding proposal, IRB or other ethics committee submission, study protocol, or other planning document written before analyzing the data) was used in designing the study, please include the relevant document(s) with your revised manuscript as a Supporting Information file to be published alongside your study and cite it in the Methods section. A legend for this file should be included at the end of your manuscript. If no such document exists, please make sure that the Methods section transparently describes when analyses were planned, and when/why any data-driven changes to analyses took place. Changes in the analysis, including those made in response to peer review comments, should be identified as such in the Methods section of the paper, with rationale.

---

## [Decision Letter · Decision Letter 2]

9 Jul 2025

Dear Dr Sheppard,

Many thanks for submitting a revision of your manuscript "Associations between falls and other serious adverse events and antihypertensive medication in individuals with dementia: An observational cohort study" (PMEDICINE-D-25-00036R2) to PLOS Medicine. The paper has been re-reviewed by two subject experts and a statistician; their comments are included below and can also be accessed here: [LINK]

As you will see, the reviewers appreciate the revisions, but have some additional concerns. After discussing the paper with the editorial team and an academic editor with relevant expertise, I'm pleased to invite you to revise the paper in response to the reviewers' comments. Specifically, we urge you to include figure S7 in the main text, with the appropriate caveats. We plan to send the revised paper to some or all of the original reviewers, and we cannot provide any guarantees at this stage regarding publication. In addition, we also have some editorial requests, which you can find at the bottom of this email. Please address these, and include a point-by-point response.

We ask that you submit your revision by Jul 30 2025 11:59PM. However, if this deadline is not feasible, please contact me by email, and we can discuss a suitable alternative.

Don't hesitate to contact me directly with any questions (sbruijn@plos.org). 

Best regards, 

Suzanne 

Suzanne De Bruijn, PhD 

Associate Editor

PLOS Medicine

sbruijn@plos.org

Comments from the academic editor:

The rebuttal is comprehensive and compelling. Although I understand the author's reservations about the interpretation of figure S7, it could be included in the main text with the appropriate caveats. 

Comments from the reviewers: 

Reviewer #2: Authors have adequately and thoroughly all raised issues. The only minor point I feel they should still modify is the comparison with metaanalysis of RCTs (lines 527-531). In fact, the metaanalysis included for comparison in their previous work (ref. 9) is indeed ref. 72, which therefore is not "another previous metaanalysis". Therefore I would simply state that the association between antihypertensive treatment and falls was greater than the estimate obtained in a meta-analysis of randomised controlled trials. In fact it is unsurprising that the results of this paper were similar to their previous work, which included the same individuals as the present study and actually shows the same difference. Auuhors might emphasize again that one likely reason, beyond residual confounding, is the already cited exclusion of frailer patients, including people with dementia, from RCTs.

Reviewer #3: Thanks for the revised manuscript and responses to my original review. There are three key changes that need attention.

I agree with reviewer 1 that the interaction tests in Table S7 are the key result if the main aim is to compare differences in risk of AE from anti-hypertensive agents according to dementia risk. Comparing the 'significance' of each the outcome-exposure relationship between no dementia/dementia patients is termed 'differences in nominal significance' which sharply increases Type I error. This can be seen in the overlapping 95% CI in the effect estimates between no dementia/dementia participants. The interactions tested in S7 are the appropriate approach to examining heterogeneity by dementia status.

It is possible to see overall similar SMDs between the groups after matching/weighting a PS where there is limited overlap or strong variation in distribution between the treated and non-treated groups. This is a relatively simple check (histograms of PS by treatment status) and should be included in a revised manuscript. 

The details of how the 'tests for trend' were completed should be included in the methods - there is a huge amount of variation in what this term means in the literature. 

Reviewer #4: Reviewer 4, 

I would like to thank the authors for their extensive replies and edits to the original manuscript. I have a few remaining queries, but nothing major.

Response to Q3: "Importantly, any differences in age structure between the dementia and non-dementia groups were accounted for in our analyses, where we adjusted for age in all models and further controlled for baseline cardiovascular and frailty-related comorbidities."

For this to work, the adjustments in the model would need to be very accurate. This is questionable, both because the relationship between e.g. age and the outcomes may be complex (non-linear), and because the authors did not adjust for time varying covariates, meaning that any incident comorbidity (which is more likely to occur in older patients) is not adjusted for. To alleviate this concern, I think a sensitivity analysis wherein individuals with dementia are more exactly matched to their age- and comorbidity peers, and subsequently assessing significant differences in the effects of antihypertensive drugs, would be prudent. 

Response to Q5: "We acknowledge the potential limitations in diagnostic accuracy and date-stamping in electronic health record databases. However, CPRD GOLD is a high-quality, widely validated UK primary care database, and previous validation studies have shown acceptable levels of diagnostic accuracy for chronic conditions, including cardiovascular disease and dementia.a,b,c Moreover, our analyses did not rely solely on diagnostic codes. We also incorporated prescription and measurement data, which are generally well recorded in CPRD. Notably, all issued prescriptions are automatically documented in the electronic health record, providing a reliable source of exposure information for phamacoepidemiological studies

The provided references (a,b,c) report sensitivity ranges of about 75% (for stroke) or only reported "positive predictive value" for dementia. Assuming an average sensitivity of 70-80%, there is a good chance that both dementia and comorbidity diagnosis may be missed, which may affect results. I think this should be mentioned in the limitations.

---

* Please upload any figures associated with your paper as individual TIF or EPS files with 300dpi resolution at resubmission; please read our figure guidelines for more information on our requirements: http://journals.plos.org/plosmedicine/s/figures. While revising your submission, please upload your figure files to the PACE digital diagnostic tool, https://pacev2.apexcovantage.com/. PACE helps ensure that figures meet PLOS requirements. To use PACE, you must first register as a user. Then, login and navigate to the UPLOAD tab, where you will find detailed instructions on how to use the tool. If you encounter any issues or have any questions when using PACE, please email us at PLOSMedicine@plos.org.

FIGURES AND TABLES

REFERENCES

OBSERVATIONAL STUDIES

* Please state in the Methods section whether the study had a prospective protocol or analysis plan. If a prospective analysis plan (from your funding proposal, IRB or other ethics committee submission, study protocol, or other planning document written before analyzing the data) was used in designing the study, please include the relevant document(s) with your revised manuscript as a Supporting Information file to be published alongside your study and cite it in the Methods section. A legend for this file should be included at the end of your manuscript. If no such document exists, please make sure that the Methods section transparently describes when analyses were planned, and when/why any data-driven changes to analyses took place. Changes in the analysis, including those made in response to peer review comments, should be identified as such in the Methods section of the paper, with rationale.

---

## [Decision Letter · Decision Letter 3]

12 Aug 2025

Dear Dr. Sheppard,

Thank you very much for re-submitting your manuscript "Associations between falls and other serious adverse events and antihypertensive medication in individuals with dementia: An observational cohort study" (PMEDICINE-D-25-00036R3) for review by PLOS Medicine.

I have discussed the paper with my colleagues and the academic editor and it was also seen again by one reviewer. I am pleased to say that provided the remaining editorial and production issues are dealt with we are planning to accept the paper for publication in the journal.

********

We look forward to receiving the revised manuscript by Aug 19 2025 11:59PM.   

Sincerely,

Suzanne De Bruijn, PhD

Associate Editor 

PLOS Medicine

plosmedicine.org

Requests from Editors:

GENERAL EDITORIAL REQUESTS

* We appreciate that you provide the text under 'summary boxes. However at PLOS we ask for a short, non-technical Author Summary of your research to make findings accessible to a wide audience that includes both scientists and non-scientists. The Author Summary should immediately follow the Abstract in your revised manuscript. This text is subject to editorial change and should be distinct from the scientific abstract. Ideally each sub-heading should contain 2-3 single sentence, concise bullet points containing the most salient points from your study. In the final bullet point of ‘What Do These Findings Mean?’ Please include the main limitations of the study in non-technical language.

Please see our author guidelines for more information: https://journals.plos.org/plosmedicine/s/revising-your-manuscript#loc-author-summary.

* In the author summary, in the final bullet point of 'What Do These Findings Mean?', please include the main limitations of the study in non-technical language.

* Please confirm that your abstract complies with our requirements, including format (three sections: Background, Methods and Findings, and Conclusions) and providing all the information relevant to this study type https://journals.plos.org/plosmedicine/s/submission-guidelines#loc-abstract

* Please ensure that all abbreviations are defined at first use throughout the text.

* Please confirm that all numbers presented in the abstract are present and identical to numbers presented in the main manuscript text.

GENERAL

* Please review your text for claims of novelty or primacy (e.g. 'for the first time') and remove this language. In addition, please check that any use of statistical terms (such as trend or significant) are supported by the data, and if not please remove them.

* Please remove the 'conclusions' subheading from the discussion. Please also remove any other subheadings from the discussion.

* Statistical reporting: Please revise throughout the manuscript, including tables and figures.

- Please report statistical information as follows to improve clarity for the reader ""22% (95% CI [13,28]; p</=)"".

- Please separate upper and lower bounds with commas instead of hyphens as the latter can be confused with reporting of negative values.

- Please repeat statistical definitions (HR, CI etc.) for each set of parentheses.

* In the abstract, please include the important dependent variables that are adjusted for in the analyses.

FUNDING STATEMENT

* The funding statement should include URLs of the funding agencies. 

COMPETING INTERESTS STATEMENT

* All authors must declare their relevant competing interests per the PLOS policy, which can be seen here: https://journals.plos.org/plosmedicine/s/competing-interests For authors with ties to industry, please indicate whether any of the interests has a financial stake in the results of the current study.

DATA AVAILABILITY

*Please provide an URL or an email address for the CPRD.

*Please include the links to NHS digital and the office for national statistics (which are mentioned in the data availability in the methods section), in the Data availability statement in the meta-data.

FIGURES

* When a p value is given, please specify the statistical test used to determine it in the legend.

* Please consider avoiding the use of red and green in order to make your figure more accessible.

OBSERVATIONAL, COHORT, CROSS-SECTIONAL, AND CASE CONTROL STUDIES

* Did your study have a prospective protocol or analysis plan? Please state this (either way) early in the Methods section.

RESULTS

*Please consider stating the primary and secondary outcomes in the subheaders, rather than just ‘primary outcome’ for ease of the reader.

Comments from Reviewers:

Reviewer #3: Thank you for the revised manuscript, this version resolves the queries from my last review. This is an interesting study, and I enjoyed reading it.

********

---

## [Editor Report · Decision Letter 4]

22 Aug 2025

Dear Dr Sheppard, 

On behalf of my colleagues and the Academic Editor, Carol Brayne, I am pleased to inform you that we have agreed to publish your manuscript "Associations between falls and other serious adverse events and antihypertensive medication in individuals with dementia: An observational cohort study" (PMEDICINE-D-25-00036R4) in PLOS Medicine.

Before your manuscript can be formally accepted, we have a few remaining requests, as previously mentioned by email:

1) Kindly change the colours of any figures that include red-green.

2) Please change the title from the authors summary from 'Summary box' into 'Author Summary'.

3) The sentence about limitations is identical in the Abstract and the authors summary; Please consider modifying the sentence in the Author summary.

4) I appreciate your response to our question about 95% CIs “Please separate upper and lower bounds with commas instead of hyphens as the latter can be confused with reporting of negative values”. Although I appreciate that your response changing the hyphen to ‘to’ makes it clearer, our house style requires a comma between the upper and lower bounds. I kindly ask you to adjust these.

Furthermore, you will need to complete some formatting changes, which you will receive in a follow up email. Please be aware that it may take several days for you to receive this email; during this time no action is required by you. Once you have received these formatting requests, please note that your manuscript will not be scheduled for publication until you have made the required changes.

PRESS

Sincerely, 

Suzanne De Bruijn, PhD 

Associate Editor 

PLOS Medicine